**EMBO** *reports*

# Multiomics analysis identifies novel facilitators of human dopaminergic neuron differentiation

Borja Gomez Ramos [1,2], Jochen Ohnmacht [1,2], Nikola de Lange[2], Elena Valceschini [1], Aurélien Ginolhac [1], Marie Catillon[1], Daniele Ferrante[2], Aleksandar Rakovic[3], Rashi Halder [2], François Massart [2], Giuseppe Arena [2], Paul Antony[2], Silvia Bolognin[2], Christine Klein[3], Roland Krause [2], Marcel H Schulz [4,5,6], Thomas Sauter [1], Rejko Krüger [2,7,8] & Lasse Sinkkonen [1✉]

## Abstract

**Midbrain dopaminergic neurons (mDANs) control voluntary movement, cognition, and reward behavior under physiological conditions and are implicated in human diseases such as Parkinson's disease (PD). Many transcription factors (TFs) controlling human mDAN differentiation during development have been described, but much of the regulatory landscape remains undefined. Using a tyrosine hydroxylase (TH) human iPSC reporter line, we here generate time series transcriptomic and epigenomic profiles of purified mDANs during differentiation. Integrative analysis predicts novel regulators of mDAN differentiation and super-enhancers are used to identify key TFs. We find LBX1, NHLH1 and NR2F1/2 to promote mDAN differentiation and show that over-expression of either LBX1 or NHLH1 can also improve mDAN specification. A more detailed investigation of TF targets reveals that NHLH1 promotes the induction of neuronal miR-124, LBX1 regulates cholesterol biosynthesis, and NR2F1/2 controls neuronal activity.**

**Keywords** Dopaminergic Neurons; Multi-Omics Data Integration; Chromatin; Enhancers; Transcription Factors
**Subject Categories** Development; Neuroscience

## Introduction

Induced pluripotent stem cell (iPSC) technology presents a unique system to study how transcription factors (TFs) control cell differentiation and specification in humans. TFs bind to genomic regulatory regions such as enhancers and promoters to mediate their action. The TFs occupying an enhancer region collectively control transcriptional initiation of their target genes through different mechanisms (Levine, 2010; Spitz and Furlong, 2012). In particular super-enhancers (SEs), dense clusters of enhancers under high regulatory load (Whyte et al, 2013), have been associated with cell identity genes, master TFs, and are enriched with disease-associated genetic variants (Hnisz et al, 2013; Parker et al, 2013). Identifying TF binding profiles and active enhancer regions across the genome requires laborious and indirect methods such as chromatin immunoprecipitation sequencing (ChIP-seq). The presence of regulatory proteins alone does not necessarily indicate active gene regulation. To overcome this limitation, these methods can be combined with gene expression analysis in an integrative multi-omics approach to determine the role of specific TFs or enhancers in gene regulation. Moreover, a time course analysis combining transcriptomic and epigenomic data has shown great potential for studying processes like cell differentiation and revealing the regulatory events' hierarchy. Such approaches have been able to generate gene regulatory networks (GRN) that also take into account the regulatory landscape of the cells, facilitating the identification of key TFs (Duren et al, 2020; Rauch et al, 2019; Gérard et al, 2018).

Midbrain dopaminergic neurons (mDANs) are widely used in biomedical research due to their involvement in different psychiatric and neurological disorders such as schizophrenia, drug addiction and PD (Volkow et al, 2017; Grace and Gomes, 2019; Okano and Morimoto, 2022). The current protocols for generating mDANs from iPSCs produce heterogeneous populations and incompletely specified cells (Fernandes et al, 2020; La Manno et al, 2016). The genetic program underlying mDAN development has been extensively investigated (reviewed by Arenas et al, 2015), with most of the studies relying on transcriptomic data and mice as the model organism. With the emergence of single-cell technologies, improved insights into human and mouse midbrain development have revealed differences in the temporal dynamics, cell composition, and expression of TFs (Ásgrímsdóttir and Arenas, 2020; La Manno et al, 2016; Kamath et al, 2022).

[1]Department of Life Sciences and Medicine (DLSM), University of Luxembourg, L-4362 Belvaux, Luxembourg. [2]Luxembourg Centre for Systems Biomedicine (LCSB), University of Luxembourg, L-4362 Belvaux, Luxembourg. [3]Institute of Neurogenetics, University of Lübeck, 23538 Lübeck, Germany. [4]Institute for Cardiovascular Regeneration, Goethe University, 60590 Frankfurt, Germany. [5]German Centre for Cardiovascular Research, Partner site Rhein-Main, 60590 Frankfurt am Main, Germany. [6]Cardio-Pulmonary Institute, Goethe University, Frankfurt am Main, Germany. [7]Centre Hospitalier de Luxembourg (CHL), L-1210 Luxembourg, Luxembourg. [8]Luxembourg Institute of Health (LIH), L-1445 Luxembourg, Luxembourg. ✉E-mail: lasse.sinkkonen@uni.lu

All these findings highlight our limited understanding of human development, hampering our ability to apply the developmental knowledge to improve iPSC differentiation protocols. Only a few studies have considered the epigenetic landscape of mDANs during development (Xia et al, 2017; Fernández-Santiago et al, 2015; Meléndez-Ramírez et al, 2021). Furthermore, cellular heterogeneity present in the iPSC-derived cultures obscure physiological or biological insights from the cell type of interest (Rakovic et al, 2022; Sandor et al, 2017).

In this study, we profiled differentiating human iPSC-derived mDANs at the transcriptomic and epigenomic levels. Integrative analysis using our EPIC-DREM pipeline (Li et al, 2019; Schmidt et al, 2017; Schulz et al, 2012; Gérard et al, 2018) generated time-point-specific gene regulatory interactions. Together with mapping of cell type-specific SEs, this allowed the identification of putative key TFs controlling mDANs. We show that LBX1, NHLH1 and NR2F1/2 are necessary for mDAN differentiation, with LBX1 and NHLH1 also able to increase the number of mDANs. Further characterization of these TFs revealed the control of cholesterol biosynthesis by LBX1, induction of miR-124 by NHLH1, and regulation of neuronal activity by NR2F1/2 as few of the mechanisms contributing to mDAN specification. In summary, this study provides novel profiling of differentiating mDANs. Our data can be exploited for further purposes such as studies on disease-associated regulatory genetic variation.

# Results

## Generation of paired transcriptomic and chromatin accessibility profiles of mDAN differentiation

mDANs were enriched for transcriptomic and epigenomic profiling based on the expression of P2A-mCherry reporter, stably expressed under the control of the endogenous promoter of the TH gene, coding for tyrosine hydroxylase, the rate-limiting enzyme for dopamine biosynthesis (Fig. 1A; Rakovic et al, 2022). The presence of the mCherry reporter allowed the purification of mDANs from heterogeneous iPSC-derived neuronal cultures by FACS at multiple time points of differentiation (Fig. 1A,B). Reporter expression correlated with TH expression, as validated by immunocytochemistry (Fig. 1B). Using the reporter cell line (TH-Rep1), paired transcriptomic and chromatin accessibility profiles were generated from neuronal progenitor cells (smNPCs) and mDANs after 15, 30, and 50 days of differentiation, as shown in Fig. 1A. Time points were selected based on culture features reported by (Reinhardt et al, 2013): mDANs appear in the culture by day 15, after the cells are incubated in maturation medium (Fig. 1A); they start to show electrophysiological activity after 30 days of differentiation (Reinhardt et al, 2013). Finally, by day 50 mDANs begin to resemble more mature neurons. Gene expression and chromatin accessibility profiles were generated from iPSC-derived astrocytes differentiated from the same reporter cell line for 65 days as additional reference data. Moreover, to control for variation between independent iPSC lines, an additional P2A-mCherry reporter line was generated and used for RNA-seq analysis of smNPCs and mDANs after 15 and 30 days of differentiation (Figure EV1).

The data generated from mDANs showed cell-type-specific gene expression and chromatin accessibility profiles based on established markers of these cells (TH, EN1, DDC, and LMX1B) when compared to non-mDANs, and in contrast to either pluripotency/early neuroectoderm markers (PAX6) or glial markers (GFAP) (Fig. 1C; Arenas et al, 2015; Nakamura and Watanabe, 2005; Yang and Wang, 2015). The transcriptome analysis of the mCherry population across differentiation confirmed a downregulation of pluripotency genes in parallel with induction of pan-neuronal markers and, importantly, mDAN-specific marker genes (Figure EV1A; Anderegg et al, 2015; La Manno et al, 2016). Most of the markers were similarly expressed in the second reporter line (Figure EV1B). Moreover, some of the genes selective for either A9 and A10 mDAN subtypes became upregulated, while others like GRP remained undetected in one or the other reporter line, suggesting that the cells have not adopted a subtype-specific identity. Principal component analysis (PCA) confirmed a clear separation of smNPCs, mDANs and astrocytes at the level of both transcriptome and chromatin accessibility (Fig. 1D). Taken together, our reporter cell lines, and the obtained genome-wide profiles, can be used for further characterization of the regulatory landscape of mDAN differentiation.

## Integrative analysis predicts key regulators of human mDAN differentiation

To integrate the paired transcriptomic and epigenomic profiles to identify key regulators of mDAN differentiation, our previously published EPIC-DREM pipeline was adapted for use with ATAC-seq data and applied (Fig. 1E; Gérard et al, 2018). In short, genome-wide accessible regions were determined per cell type and time point, and footprinting analysis was performed to predict TF binding sites (TFBS) in these regions using HINT-ATAC (Li et al, 2019). Computed TF-gene scores based on accessibility signal and TF binding strength were integrated to finally define regulators across time in previously described statistical framework (Schmidt et al, 2017, 2019). The time point-specific predictions were combined with the time series gene expression changes by DREM (Schulz et al, 2012) to build a time point-specific gene regulatory network of mDAN differentiation. DREM identifies gene expression bifurcation points across the time points analyzed, corresponding to groups of co-expressed genes. For each bifurcation point (also called split node), TFs are ranked according to the highest number of target genes within the group based on accessibility data predictions. Therefore, DREM highlights the TFs controlling most of the observed transcriptional changes across time.

Figure 2A shows the result of EPIC-DREM on mDAN time course data. There are a total of 26 split nodes with a total of 327 TFs ranked differently across them (Dataset EV1). Highlighted in red are different top-ranked TFs known to be involved in the regionalization, differentiation, and specification of mDANs. For example, well-described TFs controlling mDAN differentiation, such as NR4A2 (also known as NURR1), LMX1A/B and EN1, were identified among the main regulators (Hermanson et al, 2003; Veenvliet et al, 2013; Sherf et al, 2015; Hoekstra et al, 2013). Pioneer TFs, critical for cell reprogramming due to their ability to alter chromatin structure (Iwafuchi-Doi and Zaret, 2016), such as ASCL1 and NEUROG2, also appeared as top-ranked TFs (Herdy et al, 2019; Smith et al, 2016). Other factors more involved in the regionalization and differentiation of the midbrain such as GBX2, NKX6-1, PBX1, SREBF1, NR1H2 (also known as LXRβ), and MSX1

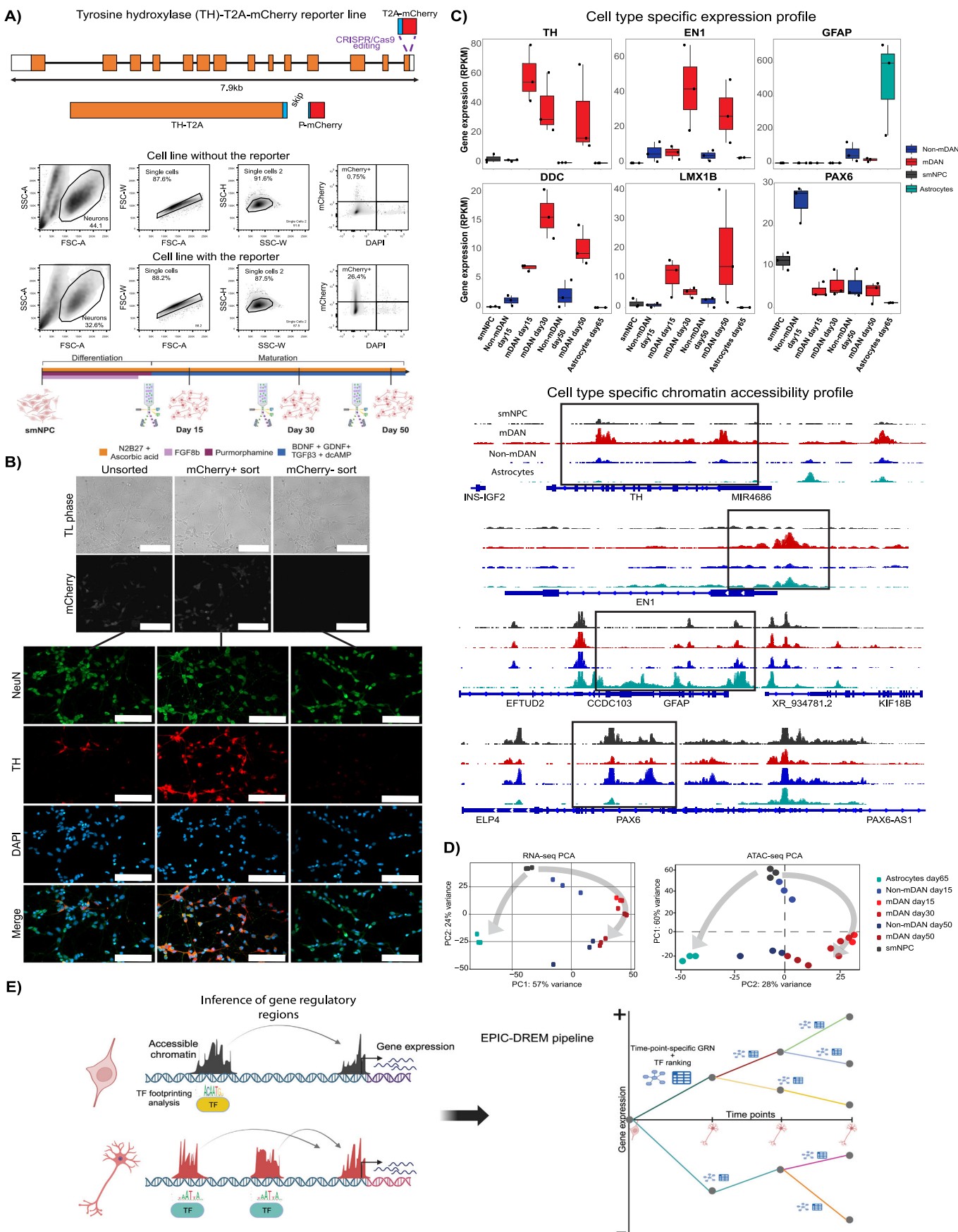

◀ **Figure 1. Epigenomic and transcriptomic analysis of human mDAN differentiation.**

(A) Gene editing scheme of the human iPSC reporter line, a representative flow cytometry comparison of iPSC-derived neurons from the lines with and without the reporter, and the differentiation protocol used with the different time points analyzed. (B) Validation of the reporter line by immunocytochemistry. TH staining correlated with mCherry signal. NeuN was used as a neuronal marker while DAPI stained all nuclei. Scale bar = 100 μm. (C) Gene expression and chromatin accessibility profiles from the different samples analyzed showed specific enrichments for cell-type-specific markers of mDANs (TH, EN1, DDC, LMX1B), astrocytes (GFAP), and non-mDANs (PAX6). ATAC-seq tracks are plotted under the same scale for comparison purposes. $N = 3$ independent experiments for all datasets and samples. In each boxplot the boxes and inner line represent the 25, 50 and 75% quantile of values. Whiskers mark the upper and lower limits of 1.5× the Inter-Quartile-Range. (D) PCA plots of the RNA-seq and ATAC-seq data reveal the differentiation dynamics. (E) EPIC-DREM pipeline scheme showing the different steps in data integration for the generation of time point-specific gene regulatory networks across differentiation. ATAC-seq was used to define gene regulatory regions and the TFs associated to them were identified through footprinting analysis. Integration with the transcriptional changes defined by RNA-seq allowed the creation of sets of co-expressed genes associated with a ranking of TFs controlling them across time. Source data are available online for this figure.

were captured by EPIC-DREM (Andersson et al, 2006; Martinez-Barbera et al, 2001; Toledo et al, 2020; Villaescusa et al, 2016; Sacchetti et al, 2009). Thus, our EPIC-DREM predictions are in line with the existing literature.

Moreover, pathway enrichment analysis of the genes included in the first four split nodes revealed enrichment for neuronal functions such as axon development and synapse formation among the upregulated genes (Fig. 2B, nodes 1 and 2) and enrichment for ribosomal RNA production and cell division for downregulated genes (Fig. 2B, nodes 3 and 4, respectively), consistent with switching from the multipotent state to neuronal differentiation. Thus, EPIC-DREM can highlight the main biological processes governing mDAN differentiation.

## Identification of key TFs controlled by super-enhancers

While EPIC-DREM allowed us to predict the TFs controlling the highest number of target genes during mDAN differentiation, we set out to further prioritize these TFs by identifying those that are themselves under high regulatory load in a cell type-selective manner (Galhardo et al, 2015; Hnisz et al, 2013). To do this, we performed low input ChIP-seq for H3K27ac, a histone mark associated with active enhancers, for mDANs at days 30 and 50 of differentiation to determine which TFs display association with SEs at the respective time point. For the association, SE regions were overlapped with the genomic coordinates of the genes encoding for TFs and expressed in a time point-specific manner to obtain a list of 49 TFs controlled by SEs across both time points (Fig. 3A, Dataset EV2). Finally, the list of all SE-associated TFs at either time point was compared with a list of TFs combining the top 20 ranked TFs from each of the split nodes from EPIC-DREM (Fig. 3B, Dataset EV2). Among the 49 SE-associated TFs, 17 were also among the top 20 ranked TFs from EPIC-DREM (Figs. 3B and EV2, Dataset EV2).

The list of 17 TFs was further explored to select the most promising candidates for functional analysis. For this, a literature search was performed to see whether these TFs had already been associated with mDAN function or development. TCF4, MEIS1, MEIS2, FOS, JUND, RARA, SP3, and TP53 have already been associated to mDAN subset specification, dopaminergic system formation, mDAN differentiation, and regulation of mDAN specific genes, and consequently were not included in follow-up experiments (Lyu et al, 2020; Mesman et al, 2021; Jiang et al, 2015; Wang and Bannon, 2005; Podleśny-Drabiniok et al, 2017; Agoston et al, 2014; Engele and Schilling, 1996). From the remaining nine TFs, HOXB3 was excluded since the accessibility at the locus in mDANs was centered on the adjacent HOXB2 instead (Figure EV2). Similarly, RFX2 and HSF4 were excluded due to low expression

and relatively narrow SE signal, respectively. The six remaining TFs, namely HOXB2, LBX1, NHLH1, NR2F1 (also known as COUP-TFI), NR2F2 (also known as COUP-TFII) and SOX4 were found to harbor the strongest accessibility and acetylation signals and, therefore, were selected for functional analysis as novel candidate regulators of mDAN differentiation (Figs. 3C and EV2).

Among the candidates, HOXB2 has been mainly associated with neural crest and hindbrain patterning (Parker et al, 2019). No relation of HOXB2 with mDANs has been described before. HOXB2 is clustered with other HOXB genes that are also proximal to the same SE. Most of the chromatin accessibility is observed within the HOXB2 gene body (Fig. 3C). HOXB2 was abundantly expressed in smNPCs and early time points of differentiation in both reporter lines but decreased in day 50 mDANs (Figs. 3C and EV3A), in keeping with its predicted role as a regulator of downregulated genes (Fig. 2A, Dataset EV1).

LBX1 has been implicated in the development of different types of neurons such as retrotrapezoid nucleus (RTN) neurons, GABAergic neurons, and somatosensory interneurons in the spinal cord (Gross et al, 2002; Hernandez-Miranda et al, 2018; Huang et al, 2008), highlighting a role for this TF in cell fate decisions. LBX1 was the only TF among the selected candidates to present an mDAN-specific SE (Fig. 3C). Consistently, LBX1 expression was induced >10-fold upon early mDAN differentiation in the first reporter line. However, in the second reporter line no upregulation was observed at the measured time points (Figure EV3A).

NHLH1 is mainly characterized as an early pan-neuronal marker that determines many neuronal cell fate decisions (Ratié et al, 2014; De Smaele et al, 2008; Krüger et al, 2004; Schmid et al, 2007). NHLH1 expression was increased during differentiation in both reporter lines with almost 70 RPKM detected in day 50 mDANs (Figs. 3C and EV3A).

Temporal expression dynamics of NR2F1 and NR2F2 in the developing brain are important for the specification and balance of different neuronal lineages as well as neuron-to-glia transitions (Naka et al, 2008; Bonzano et al, 2018; Teratani-Ota et al, 2016; Zhang et al, 2020). Recently, NR2F1 was found to be deregulated in iPSCs from PD patients carrying the LRRK2-G2019S mutation (Walter et al, 2021). During early mDAN differentiation NR2F1 expression increased to high levels, while also NR2F2 maintained high expression during differentiation (Figs. 3C and EV3A).

SOX4, like many SOX genes, has been described to be implicated in neurogenesis and maintenance of progenitor cells during development. Interestingly, the role of this TF in generating TH-expressing cells from sympathetic or enteric nervous systems has been described (Memic et al, 2018; Potzner et al, 2010). Also, SOX4

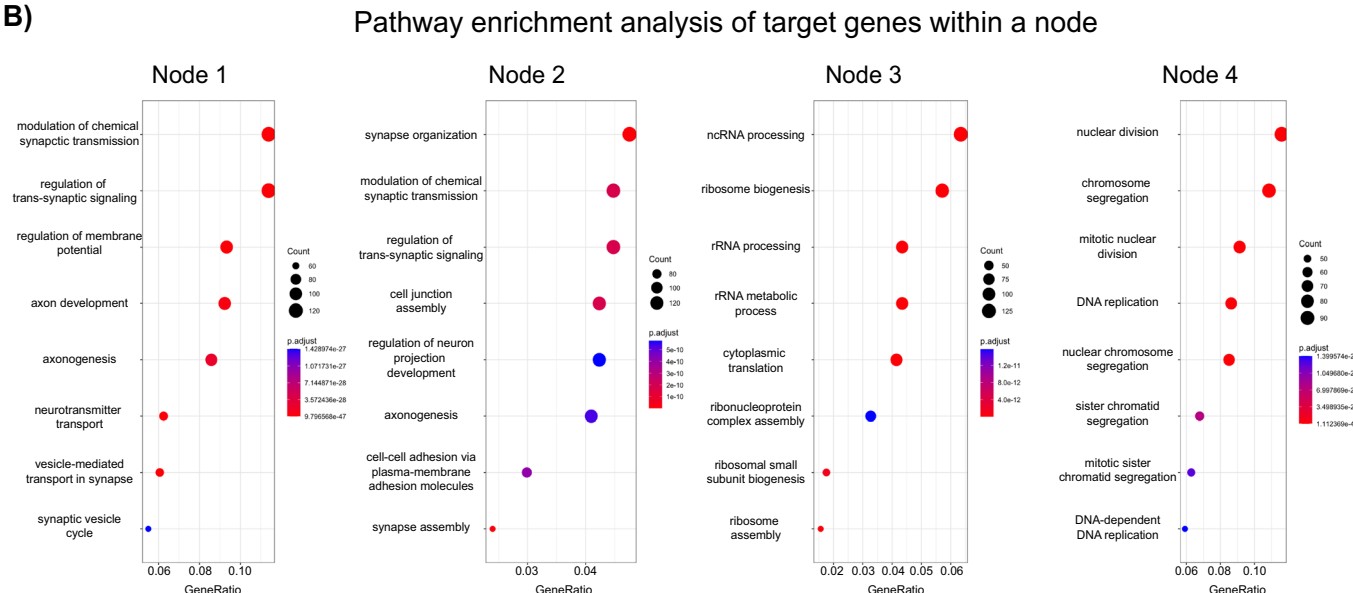

**A)**

Time course data integration by EPIC-DREM

**B)**

Pathway enrichment analysis of target genes within a node

**Figure 2.  Integrative analysis of time series transcriptomic and chromatin accessibility data predicts key regulators of mDAN differentiation.**

(A) EPIC-DREM results from the integration of time course data on gene expression and chromatin accessibility profiles of mDAN differentiation from Fig. 1. *x*-Axis represents the time points analyzed and *y*-axis represents expression log$_2$-fold changes across time. EPIC-DREM result contained a total of 26 split nodes with an associated list of TFs ranked according to their regulatory importance for the genes contained within the node. See also Dataset EV1. Highlighted in red are TFs ranked as top regulators and previously associated to control of mDAN differentiation. Highlighted in blue are the novel identified TFs that were selected for functional analysis. (B) Pathway enrichment analysis using the Fisher's exact test and hypergeometric distribution for the genes contained in the first 4 nodes created by EPIC-DREM captures the main biological processes regulated at early stages of neuronal differentiation.

expression increased to >100 RPKM during mDAN differentiation in both reporter lines and was accompanied by an increased enhancer signal at the locus (Figs. 3C and EV3A).

Since the exact expression dynamics of the candidate TFs were variable between the two TH reporter lines, especially for LBX1, we set out to confirm the induction of the TFs in additional iPSC lines. To do this we leveraged the recently published RNA-seq data of mDAN differentiation from 95 iPSC lines generated by the Parkinson's Progression Markers Initiative (PPMI) (Bressan et al, 2023). Consistently with our findings, all candidate TFs were significantly upregulated during mDAN differentiation across the studied cell lines (Figure EV3B). Overall, these candidates present great potential as cell fate and differentiation determinants with no previous relation to mDANs.

## LBX1, NHLH1, and NR2F1/2 are necessary for mDAN neurogenesis

To functionally validate the data-driven predictions in vitro, TFs were knocked down (KD) during differentiation to determine their impact on the mDAN number in mixed cultures. A lentiviral vector containing a shRNA targeting the different candidate TFs, and including a GFP reporter to control for transduction efficiency, was used. To assess the effect of the TFs, two different experimental designs for transduction were used: early and late transduction (Fig. 4A). For early transduction, cells were transduced on day one of differentiation while for late transduction, cells were transduced on day nine, just before they entered the maturation phase. In both cases, the cells were analyzed on day 15 (Fig. 4B–F). LMX1B served as a positive control as its role in mDAN differentiation is well characterized (Wever et al, 2019; Sherf et al, 2015). SOX4 was omitted as we were unable to identify an shRNA with a KD efficiency of >25%. As expected, LMX1B KD reduced mDAN numbers in the cultures as assessed by the mCherry reporter signal (Fig. 4C). Upon early transduction, LMX1B KD reduced the numbers of mDAN in the culture by ~60%. However, the KD efficiency for LMX1B, as measured by RT-qPCR, was very variable, which can be explained by the low levels of GFP positive cells (~40%) upon analysis on day 15 of differentiation (Appendix Fig. S1A), possibly masking the KD of this TF by the GFP negative cells in the culture. On the other hand, with late transduction, good transduction efficiency and strong KDs were observed on the day of analysis for LMX1B. Interestingly, late KD of this TF reduced the mDAN numbers by only 30% but also decreased the overall cell density in the cultures. This highlights the dual role of LMX1B during differentiation and correlates with previous studies (Wever et al, 2019), emphasizing the biological relevance of our cellular system.

LBX1 KD by early transduction was found to severely decrease the cell numbers, with very few cells remaining by day 5-6 post

transduction, preventing a more detailed analysis of the effect of this TF on mDANs (Appendix Fig. S1A). Although LBX1 KD by late transduction also resulted in some decrease in cell numbers, the surviving population displayed good transduction efficiencies and allowed us to study the impact of LBX1 on mDANs. After late transduction, LBX1 KD efficiency on the day of analysis was ~60% and mDAN numbers were reduced by 50% (Fig. 4B,C).

NHLH1 KD had the strongest effect on mDAN numbers among the candidates tested. KD efficiencies in both early and late transduction were around 80% and mDAN numbers were reduced by roughly 80% and 60% in early and late transduction, respectively. Although the total number of cells was not affected by the NHLH1 KD in early transduction, GFP-positive cells represented only half of the cells on the day of analysis, similar to what was observed for LMX1B KD (Appendix Fig. S1A).

Since NR2F1 and NR2F2 are known to act redundantly (Naka et al, 2008), lentiviral particles expressing the respective shRNAs were combined to target both TFs to avoid possible compensatory mechanisms. While shRNAs for both NR2F1 or NR2F2 are functional individually, in dual KD conditions only NR2F1 levels were significantly reduced (Fig. 4B). Nevertheless, the combined NR2F1/2 KDs markedly affected mDAN differentiation, reducing mDAN numbers in the culture by close to 80% and 50% in early and late transduction, respectively. In addition, when NR2F1/2 were KD, cells could keep good transduction efficiencies until the day of analysis despite the negative effect on mDAN numbers, in contrast to what was observed for LMX1B and NHLH1 KDs (Appendix Fig. S1A).

Lastly, for HOXB2 good transduction efficiencies were observed on the day of analysis, with >60% KD efficiency in both early and late transduction (Fig. 4B). However, although HOXB2 KD reduced the numbers of mDANs in the culture, the effect was the weakest among the tested TFs (Fig. 4C).

Thus, LBX1, NHLH1 and NR2F1/2 were all found to be necessary for mDAN differentiation. Their depletion during the specification of these neurons exhibited a more robust phenotype than that of LMX1B, a well-established regulator of mDAN differentiation, thereby demonstrating an important regulatory role for LBX1, NHLH1, and NR2F1/2. On the other hand, loss of HOXB2 only had a limited effect on mDAN levels.

To further confirm the role of LBX1, NHLH1, and NR2F1/2 also in additional cell lines, we performed high-content imaging of TH staining in four independent iPSC lines differentiated towards mDANs and using the late transduction protocol. More specifically, we studied two non-reporter lines (17608 and K2135) and two reporter lines (TH-Rep1 and TH-Rep2) and used MAP2 staining as a broad neuronal marker. The overall impact of the TF KDs was dependent on the level differentiation achieved in each cell line, with strongest effects observed in the 17608 line where depletion of

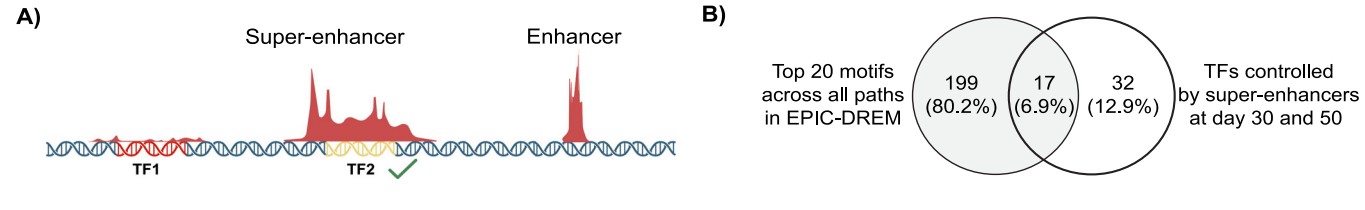

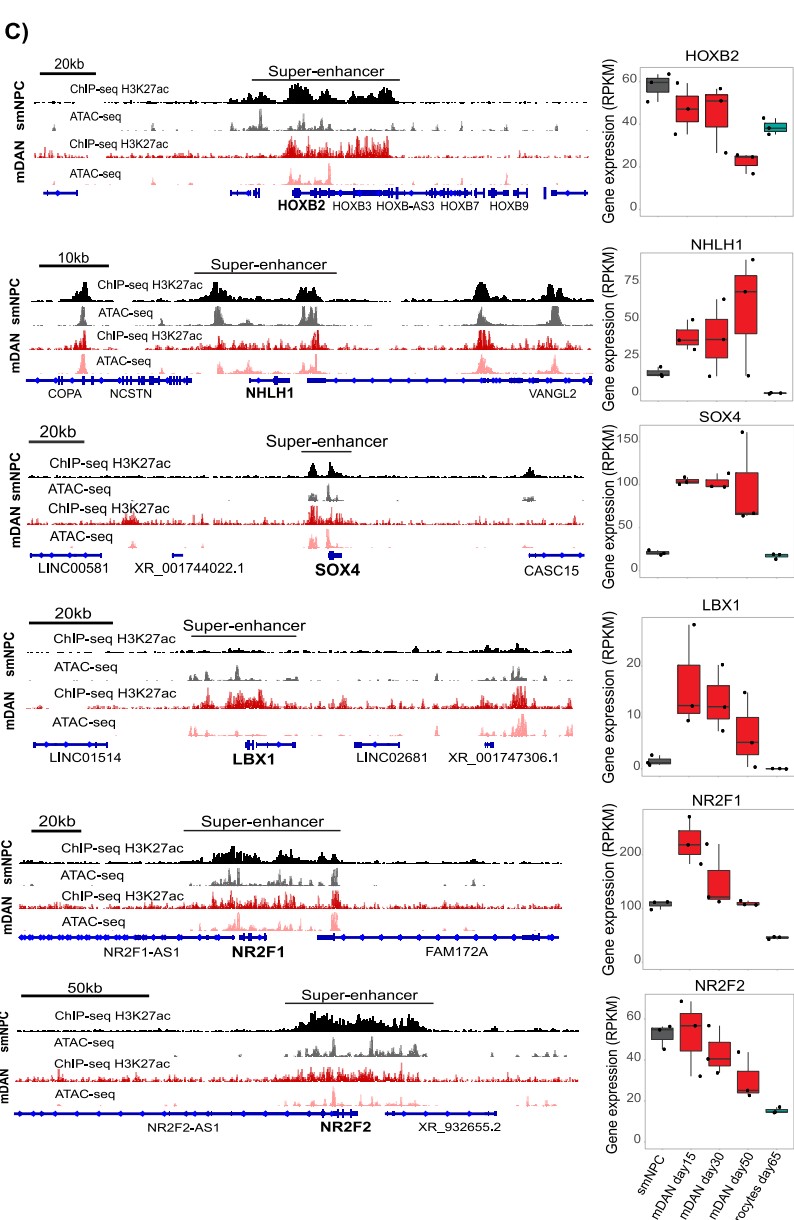

**Figure 3. Identification of key TFs controlled by super-enhancers.**

(A) Schematic representation of the selection of TFs controlled by super-enhancers. First, TF has to be expressed at the specific time point of analysis and second, its gene body had to be located under a super-enhancer region in order to be included. (B) Venn analysis of the top 20 TFs across all split nodes from EPIC-DREM with the list of TFs controlled by super-enhancers across the analyzed time points. (C) H3K27ac signal and chromatin accessibility profiles at the loci of the novel candidate TFs, highlighting the identified super-enhancer regions in mDANs, together with the expression dynamics during mDAN differentiation. ATAC-seq and ChIP-seq tracks are plotted under the same scale per dataset for comparison purposes. Data are representative of 3 independent experiments. Boxplots illustrate the distribution of data as described in Fig. 1.

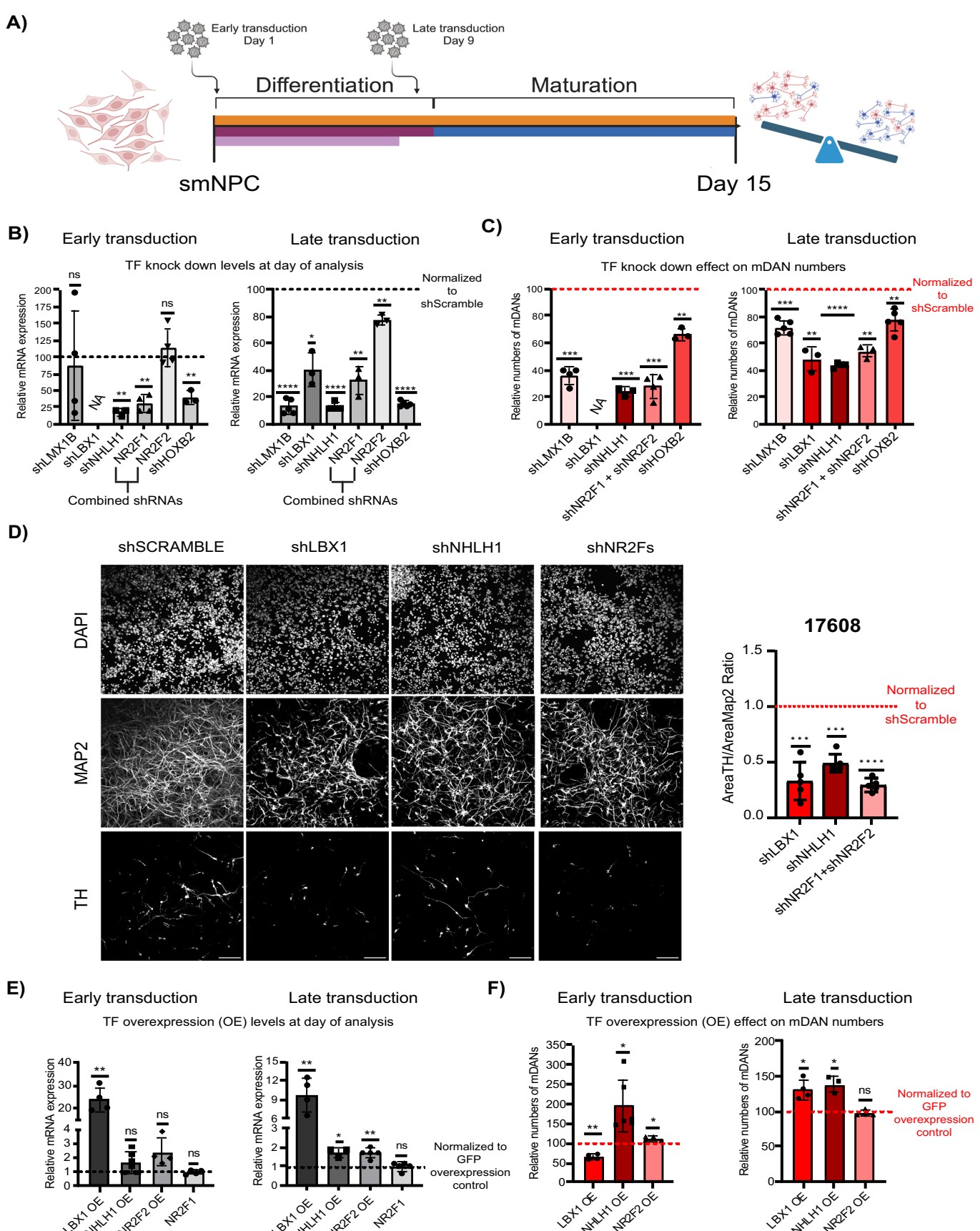

**Figure 4. LBX1, NHLH1, and NR2F1/2 are necessary for mDAN differentiation and LBX1 and NHLH1 can also improve mDAN neurogenesis.**

(A) Graphic representation of the two different transduction approaches (day 1 and day 9) used in this study and the day of analysis (day 15) for all samples. (B) TF knock down results for early and late transduction of shRNA lentiviral particles. mRNA levels were always normalized to shScramble per replicate. (C) Relative mDAN numbers were calculated based on mCherry signal and normalized to the mCherry population of the shScramble condition per replicate. (D) High-content imaging analysis of 17608 cell line at day 15 of differentiation following late transduction with shRNA lentiviral particles. Cells were stained for the nuclear marker DAPI, the neuronal marker MAP2 and TH as the marker for mDANs. Scale bar = 100 μm. Quantification of the TH-stained area over the MAP2 area are shown as bar plots. Ratios were normalized to the shSCRAMBLE per replicate. (E) TF overexpression results for early and late transduction of lentiviral particles containing codon optimized cDNA. Expression was normalized to GFP overexpression per replicate. (F) Relative mDAN numbers were also calculated according to mCherry signal but normalized to GFP overexpression per replicate. Data information: Data in all panels are representative of at least 3 independent experiments. Error bars (B, C, E, F) correspond to ±1 standard deviation (SD) from the mean. One sample *t* test was used for statistical analysis, taking 100 (C) or 1 (F) as the theoretical mean. *$p$ value <0.05, **$p$ value <0.01, ***$p$ value <0.001, ****$p$ value <0.0001, and ns not significant. Source data are available online for this figure.

each TF significantly reduced the TH area in comparison to MAP2 area by over 50% (Fig. 4D). A similar trend for reduced TH covered area was also observed with K2135 and TH-Rep1 cells (Figure EV4A,B). The effect of NHLH1 KD was confirmed to be statistically significant in K2135 cells while effects of NR2F1/2 KD in K2135 cells and LBX1 KD in TH-Rep1 cells were found to be just above the significance cut-off ($p = 0.055$ and $p = 0.070$, respectively). The independent experiments with TH-Rep2 cell line were highly variable and no reproducible conclusion regarding the effects of the TF KDs on TH covered area could be taken. The microscopy images applying signal masks for TH can be found in Appendix Fig. S2 (see "Methods" for details).

Taken together, all three TFs were found necessary for mDAN differentiation in independent iPSC lines.

## Elevated expression of LBX1 or NHLH1 can increase mDAN numbers

To further determine the role of the identified TFs in mDANs, LBX1, NHLH1 and NR2F1/2 were overexpressed during differentiation of TH-Rep1 line using the same lentiviral transduction approach as in the earlier KD experiments. Figure 4E,F shows the results from the overexpression experiment for the three different TFs. For LBX1, strong overexpression (9–20-fold compared to control vector) and high levels of transduced cells were observed on the day of analysis (Appendix Fig. S1B) for both early and late transduction. However, the induction of LBX1 had opposite effects depending on the time point of differentiation. While LBX1 overexpression early during differentiation had a negative impact on mDAN numbers, late overexpression resulted in significantly increased mDAN numbers (Fig. 4F).

Overexpression of NHLH1 during early differentiation resulted in an almost 2-fold increase in mDANs, despite a modest increase of gene expression levels by 50% and reduced number of transduced cells based on the GFP signal on the day of analysis. With late induction of NHLH1 expression, again, high levels of transduced cells, and stable overexpression could be observed, leading to a significant increase in mDAN numbers (Fig. 4E,F).

We were unable to achieve any meaningful overexpression of NR2F1, most likely due to its already very high endogenous expression levels (Fig. 3C). Therefore, only NR2F2 overexpression was tested. While the expression fold change upon NR2F2 overexpression was comparable to that achieved for NHLH1, this had almost no effect on the cultures (Fig. 4E,F). Early transduction showed a very minor, albeit significant increase in mDAN levels, while late transduction for NR2F2 overexpression had no

observable effect on mDAN numbers. Interestingly, NR2F2 overexpression could not induce NR2F1 expression (Fig. 4E).

Taken together, LBX1 and NHLH1 were able to increase the number of mDAN, with the timing of overexpression being the key to producing this effect for LBX1. NHLH1 showed the strongest effect on mDAN numbers, and the effect was independent of the developmental timing. Lastly, NR2F2 overexpression did not present any benefit regarding mDAN neurogenesis. This is likely due to the considerable levels of expression by the endogenous and functionally redundant NR2F1.

## NHLH1 controls miR-124-3p expression in mDANs

To further characterize the role of NHLH1 in mDAN differentiation, RNA-seq was performed using the TH-Rep1 samples from the late transduction KD experiments (Fig. 4A). NHLH1 KD led to a total of 491 differentially expressed genes (DEGs) (absolute log$_2$-fold change >1, FDR <0.05) (Fig. 5A, Dataset EV3). Using an Ingenuity Pathway Analysis (IPA) for upstream regulators explaining the transcriptional changes produced by NHLH1 KD (Fig. 5B, top 10 upstream regulators based on $z$-score, $p$ value <0.05) predicted miR-124-3p to be downregulated and to belong to the key regulators driving the expression changes (Krämer et al, 2014). Consistently, g:Profiler (Raudvere et al, 2019) analysis of upregulated genes from NHLH1 KD predicted miR-124-3p to control many of the genes (Dataset EV3). Moreover, miR-124-3p was the only regulator predicted by both methods, IPA and g:Profiler. miR-124 is the most abundant microRNA in the brain with neurogenic properties and has been associated with dopaminergic neurodegeneration in PD (Angelopoulou et al, 2019). Indeed, further exploration of our epigenomic data from mDANs confirmed the three loci coding for miR-124 (MIR124-1 to -3) to be highly accessible with large regions occupied by H3K27ac at MIR124-2 and -3 (Appendix Fig. S3A). Moreover, the primary transcripts of miR-124-1 and miR-124-2 presented an increased expression specifically in mDANs (Appendix Fig. S3A). To validate the predicted reduction in miR-124 expression upon NHLH1 depletion, a TaqMan microRNA assay was used. Importantly, a strong and significant downregulation of mature miR-124-3p could be confirmed (Fig. 5C).

Taking advantage of TargetScan (https://www.targetscan.org), a database containing predicted microRNA targets (Agarwal et al, 2015), a list of predicted targets for miR-124-3p was filtered for the genes also expressed in our RNA-seq data (Dataset EV3). The obtained gene list was compared with the list of upregulated DEGs from NHLH1 KD to determine how many of the genes could be

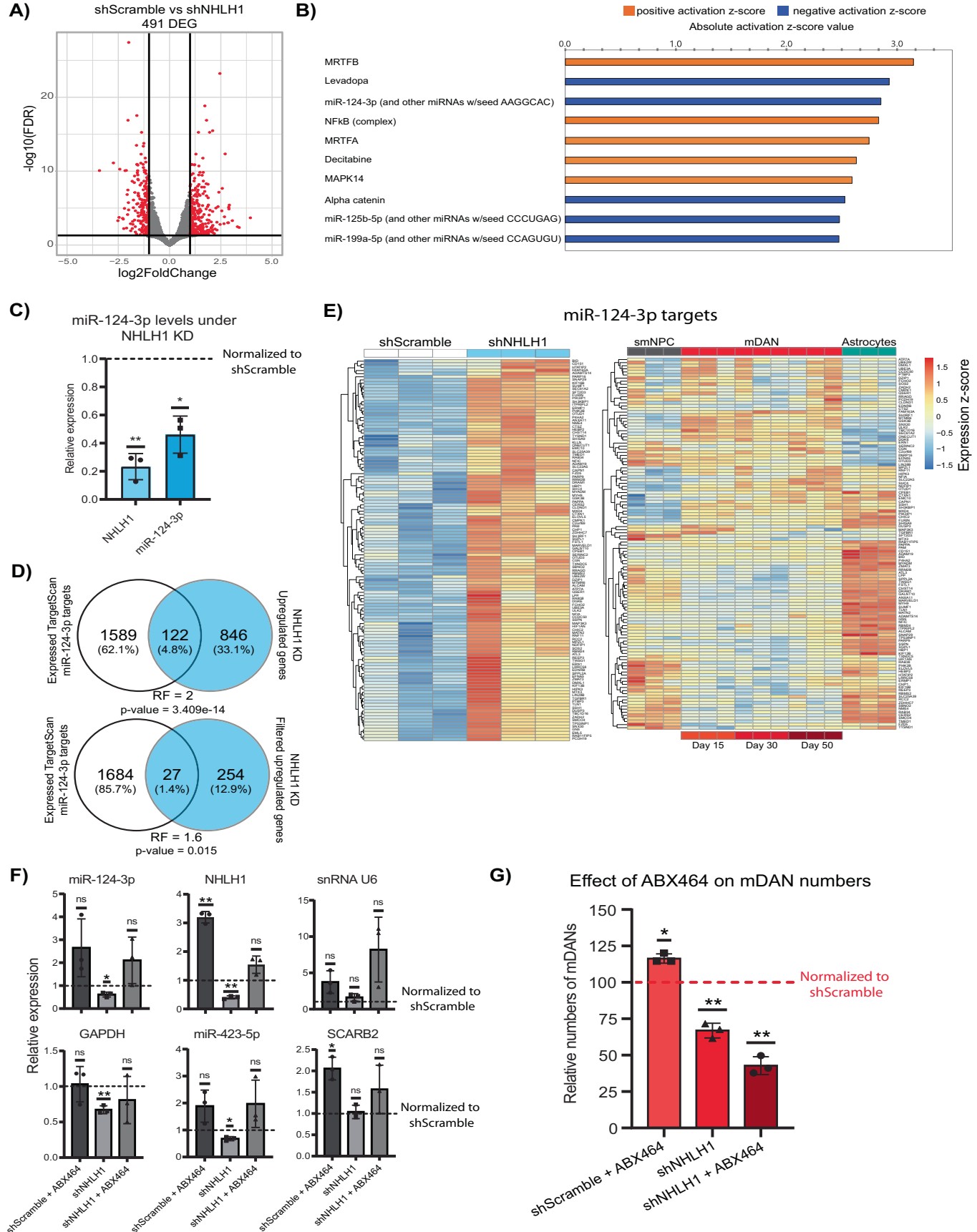

◄ **Figure 5.  NHLH1 controls mDAN differentiation by regulating miR-124-3p expression.**

(A) Volcano plot showing DEGs from RNA-seq analysis of neurons at day 15 following a late transduction with shScramble or shNHLH1. Black lines represent cut-off according to FDR <0.05 and absolute $\log_2$-fold change >1. (B) Ingenuity Pathway Analysis (IPA) predicted top 10 upstream regulators based on absolute $z$-score value using DEGs from (A). $p$ Value <0.05 for all regulators. (C) Bar plots showing results from TaqMan assay determining miR-124-3p levels upon NHLH1 KD. Expression was normalized to shScramble samples. (D) Venn analysis of predicted and expressed miR-124-3p targets from TargetScan with upregulated genes upon NHLH1 KD. Two overlaps were done, one using the upregulated genes with a $p$ value <0.05 and the second with upregulated genes with a $\log_2$-fold change >1 and $p$ value <0.05. For overlaps, a hypergeometric test was used to determine statistical significance. RF representation factor. RF >1 indicates more overlap than expected by chance. (E) Heatmaps showing the expression of the 122 predicted miR-124-3p targets upregulated upon NHLH1 KD. Genes were plotted in the RNA-seq data from the KD experiment and the time course data containing smNPC, mDANs and astrocytes. (F) RT-qPCR and TaqMan assays were used to determine the expression of different genes and microRNAs upon ABX464 treatment and NHLH1 KD conditions. Expression values were all normalized to ACTB. Normalization between groups was done using shScramble as the reference condition. (G) Effect of ABX464 treatment and NHLH1 KD conditions on mDAN numbers based on mCherry signal. mDAN numbers were normalized to shScramble. Data information: Data are representative of 3 independent experiments. Error bars (C, F, G) correspond to ±1 standard deviation (SD) from the mean. One sample $t$ test was used for statistical analysis, taking 1 (C, F) or 100 (G) as the theoretical mean. *$p$ value <0.05, **$p$ value <0.01, and ns not significant. Source data are available online for this figure.

affected by the downregulation of miR-124 (Fig. 5D). Strikingly, over 12% of all upregulated genes belonged to primary targets of miR-124, a significantly larger proportion than expected by chance (hypergeometric test, $p$ value = 3.409e−14). Plotting the 122 targets of miR-124-3p in the KD RNA-seq data confirmed a strong upregulation, as expected (Fig. 5E). When the expression of the same targets in smNPCs, across mDAN time course, and astrocytes was plotted, it became clear that most of the microRNA targets are enriched in astrocytes or smNPCs (Fig. 5E). Overall, the results highlight miR-124 as a likely mediator of NHLH1-controlled gene regulation, contributing to mDAN specification.

To directly test the contribution of miR-124, a small molecule treatment was used to stimulate miR-124 expression during differentiation and determine whether there would be any benefit for mDAN differentiation. Recent studies have described a new role for ABX464, a quinoline with antiviral properties, in the selective and specific induction of miR-124 (Tazi et al, 2021; Vautrin et al, 2019; Daien et al, 2022). This molecule binds to the Cap binding complex (CBC) at the 5' end of the primary transcript and promotes the selective splicing of LINC00599, the host gene of miR-124-1. We found that ABX464 could induce miR-124-3p by around two-fold, but the required concentration had a negative effect on cell density in culture (Appendix Fig. S3B). Nevertheless, as small molecules can be powerful tools for their use in biomedicine, and their optimization is possible by chemical modifications, we proceeded with the testing ABX464 during mDAN differentiation. ABX464 was added to the cells when they entered the maturation medium on day 9 of differentiation. The molecule was tested under normal differentiation and in cells transduced by shNHLH1.

Interestingly, ABX464 treatment strongly affected both mRNA and microRNA expression (Fig. 5F, for original Ct values, please see Appendix Fig. S3D). Moreover, ABX464 also increased U6 snRNA, preventing its use for normalization, while another endogenous microRNA (miR-423-5p) and SCARB2 mRNA were also affected. Hence, we normalized all mRNAs and microRNAs to ACTB expression. Although not significant, ABX464 treatment appeared to increase miR-124-3p levels, while it was downregulated upon NHLH1 KD, as expected. NHLH1 expression was also upregulated upon ABX464 treatment that even reversed NHLH1 repression in NHLH1 KD cells. The rescue of NHLH1 and the increase of miR-124-3p in the cells treated with both ABX464 and shNHLH1 was not enough to rescue the mDAN loss (Fig. 5G). However, ABX464 treatment alone significantly increased mDAN numbers in the

culture, possibly due to increased NHLH1 and miR-124-3p expression (Fig. 5F,G and Appendix Fig. S3D).

## LBX1 regulates cholesterol metabolism

LBX1 KD resulted in a total of 2241 DEGs as assessed by RNA-seq (Fig. 6A, Dataset EV4). Next, pathway enrichment analysis of the identified DEGs was performed using IPA to determine the main processes altered by the LBX1 KD (Krämer et al, 2014). Figure 6B shows the top 10 pathways affected by the KD based on the significance of the enrichment. Cholesterol biosynthesis appeared as one of the main pathways affected by LBX1 and was predicted to be reduced upon LBX1 depletion. Indeed, most genes involved in cholesterol biosynthesis according to the human metabolic reconstruction (RECON) (Brunk et al, 2018), were significantly downregulated upon the KD (Fig. 6C). TFs involved in cholesterol metabolism have been previously associated with mDAN development and neurogenesis (Toledo et al, 2020; Sacchetti et al, 2009). Consistently, we found the upstream transcriptional regulators of cholesterol metabolism, SREBF1 and SREBF2 (Horton et al, 2002), to be downregulated by LBX1 KD, although only SREBF1 was reduced more than twofold (Fig. 6D). The decreased levels were also confirmed by RT-qPCR (Fig. 6E). Moreover, upstream regulators of SREBF1, NR1H3 and NR1H2 (LXRα/β) were also found to be deregulated (LXRα/NR1H3 $\log_2$-fold change = −0.515 and LXRβ/NR1H2 $\log_2$-fold change = 0.87). This is consistent with the observed pathway enrichment for LXR activation (Fig. 6B). Finally, SREBF1 and NR1H2 appeared as top regulators in our EPIC-DREM analysis (Fig. 2A).

These results suggested a novel role of LBX1 in controlling mDAN differentiation via cholesterol metabolism. Therefore, stimulating the cholesterol metabolism pathway should promote mDAN differentiation and could rescue the LBX1 KD effect. GW3965 is a potent and well-described synthetic agonist of NR1H3/2 (LXRα/β). LXR activation can lead to SREBF1 induction and improved mDAN differentiation (Toledo et al, 2020). Hence, neurons were treated with GW3965 under normal differentiation or upon LBX1 KD, starting from day 9 of differentiation, and analyzed on day 15 of differentiation. Although GW3965 treatment induced a high expression of SREBF1 in comparison with control conditions, the treatment did not increase mDAN numbers and did not rescue the LBX1 KD effect (Fig. 6F,G and Appendix Fig. S4). Thereby, suggesting that the critical role of LBX1 in mDAN differentiation lies upstream of cholesterol metabolism.

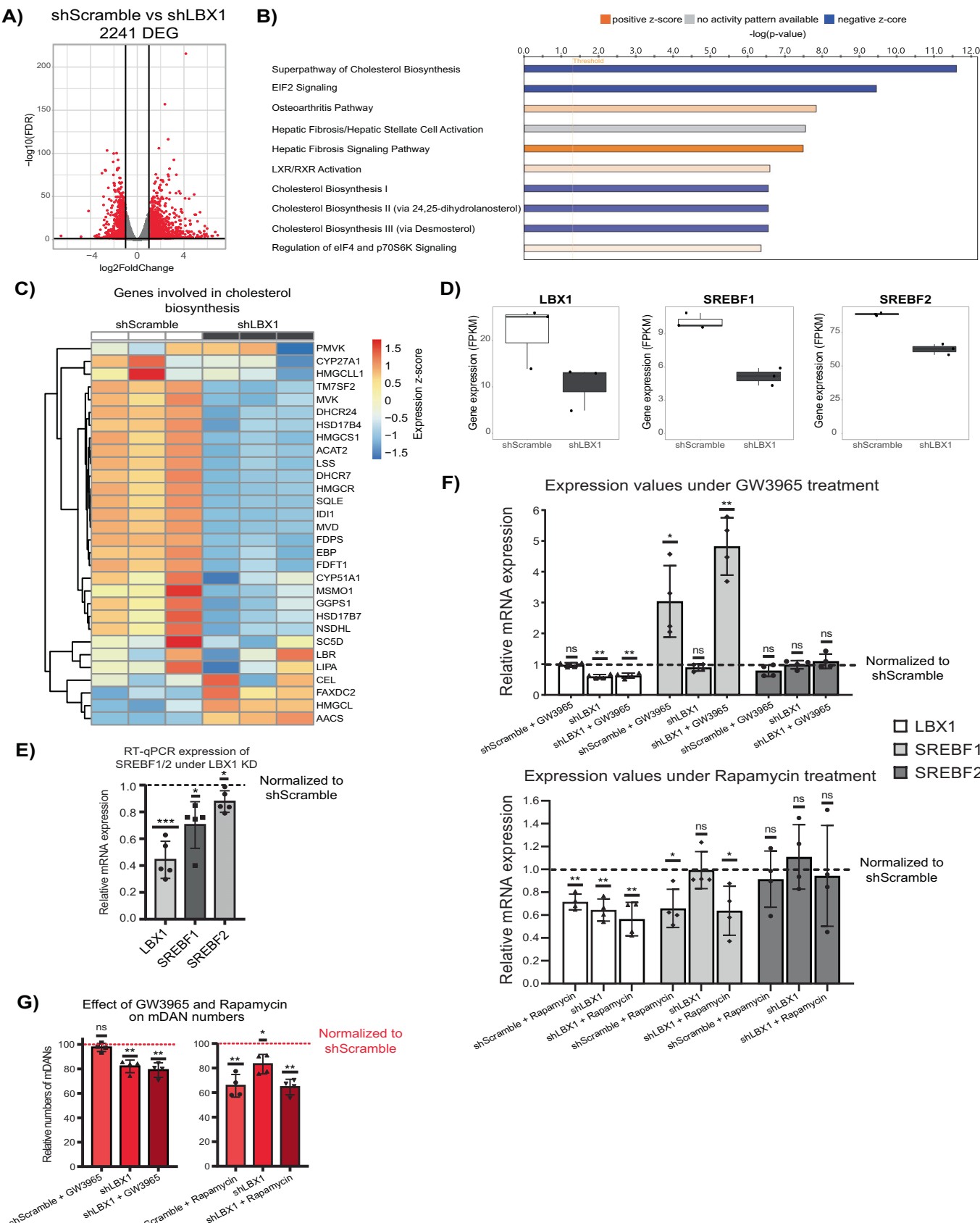

◄ **Figure 6. LBX1 controls cholesterol metabolism.**

(A) Volcano plot showing DEGs from RNA-seq analysis of neurons at day 15 following a late transduction with shScramble or shLBX1. Black lines represent cut-off according to FDR <0.05 and absolute log$_2$-fold change >1. (B) Top 10 pathways from Ingenuity Pathway Analysis (IPA) based on significance of enrichment using DEGs from (A). x-Axis represents the −log$_{10}$ p value of the enrichment. (C) Heatmap showing the differences in expression between shScramble and shLBX1 of the genes involved in cholesterol biosynthesis. (D) Expression levels (FPKM) of LBX1, SREBF1 and SREBF2 in the RNA-seq analysis. (E) RT-qPCR validation of SREBF1/2 downregulation due to LBX1 KD in late transduced neurons at day 15 of differentiation. mRNA levels were normalized to shScramble per replicate. (F) Impact of GW3965 and rapamycin treatments on gene expression in differentiating mDANs and upon LBX1 KD. Expression of LBX1, SREBF1, and SREBF2 was normalized to shScramble per replicate. (G) Effect of GW3965 and rapamycin treatments on mDAN numbers based on mCherry signal and normalized to the mCherry population of the shScramble per replicate. Data information: Data are representative of 3 (A, D), 5 (E), and 4 (F, G) independent experiments. Error bars (D–G) correspond to ±1 standard deviation (SD) from the mean. One sample t test was used for statistical analysis, taking 1 (E, F) or 100 (G) as the theoretical mean. *p value <0.05, **p value <0.01, ***p value <0.001, and ns not significant. Boxplots illustrate the distribution of data as described in Fig. 1. Source data are available online for this figure.

Further exploration of the DEGs altered by LBX1 KD high-lighted the downregulation of eIF2 signaling and alterations in the regulation of eIF4 and p70S6K signaling (Fig. 6B). Consistently, we found mTOR signaling to be predicted as downregulated upon LBX1 KD (mTOR signaling, −log(p value) = 3.97, z-score = −0.707, Dataset EV4). It has been previously shown that mTOR signaling can control lipid metabolism through SREBF1, and mTOR inhibition leads to eIF4 sequestering by the eukaryotic initiation factor 4E-binding proteins (4E-BPs), impeding translation (Wang et al, 2019; Liu and Sabatini, 2020). Therefore, downregulation of mTOR signaling due to the LBX1 KD could explain the observed downregulation in translation and cholesterol biosynthesis. Indeed, we found sirolimus (rapamycin), an mTOR inhibitor, as a chemical drug showing increased predicted activity among upstream regulators in our IPA analysis (activation z-score = 3.031, p value of overlap = 1.21e−14, Dataset EV4). Therefore, the role of mTOR signaling during mDAN differentiation was tested using rapamy-cin. Neurons were treated with rapamycin as they were treated with GW3965, under normal differentiation and under LBX1 KD. Rapamycin treatment downregulated LBX1 in cells only transduced with shScramble (Fig. 6F). In addition, rapamycin treatment was able to downregulate SREBF1 and SREBF2 in a similar way as observed before for LBX1 KD alone. Lastly, rapamycin treatment was also able to decrease the number of mDANs present in cultures (Fig. 6G). However, rapamycin did not induce a similar loss of overall cell numbers as observed upon LBX1 KD (Appendix Fig. S4).

In summary, LBX1 KD results in an extensive deregulation of the neuronal transcriptome. Cholesterol biosynthesis appeared as one of the main affected pathways regulated by LBX1, possibly through the regulation of SREBF1/2 and NR1H3/2. Stimulating this pathway using GW3965 did not result in increased numbers of mDANs, suggesting that misregulation of the TFs involved in lipid metabolism alone is insufficient to explain the observed impact on mDANs. Perturbation of mTOR signaling provides a possible explanation of the results observed upon LBX1 KD. Indeed, mTOR inhibition was able to reproduce many of the changes induced by LBX1 KD.

### NR2F1 and NR2F2 control neuronal activity

A parallel KD of both NR2F1 and NR2F2 led to misregulation of 655 genes based on RNA-seq analysis of differentiating TH-Rep1 cells (Fig. 7A, Dataset EV5). Pathway enrichment analysis of the DEGs revealed significant enrichments for processes such as synaptogenesis and axonal guidance signaling (Fig. 7B). This

suggests an involvement of NR2F1 and NR2F2 in the control of neuronal activity.

To directly investigate the electrophysiological properties of neurons upon depletion of NR2F1/2, we performed multielectrode array (MEA) recordings every five days during mDAN differentiation starting from day 15 until day 40 and following a late transduction of shRNA constructs at day 9 of differentiation (Fig. 7C). A representative raster plot of the recordings is shown in Fig. 7D. In parallel we also tested the effect of LBX1 and NHLH1 KDs as both TFs were found to be necessary for successful mDAN differentiation above (Fig. 4). The mean firing rate of the neurons was lower in both LBX1 and NHLH1 KD cells compared to control cells for most of the differentiation (p = 0.068 and p = 0.128, respectively) (Fig. 7E,F). Interestingly, the mean firing rate of neurons upon NR2F1/2 KD was particularly low, with many electrodes recording no spikes in these cells (Fig. 7G). Indicating that NR2Fs contribute to control of neuronal activity.

## Discussion

Here we present a multiomics data integration approach to discover novel TFs controlling mDAN differentiation. Although previous studies have used similar transcriptomic and epigenomic profiling of mDANs (Xia et al, 2017; Meléndez-Ramírez et al, 2021), our assessment of human mDAN differentiation is cell-type-specific and time-resolved. Together with the de novo identification of key regulators of mDANs, we also provide a functional validation and characterization of the newly identified TFs.

The EPIC-DREM pipeline offers an unbiased and data-driven tool for identifying TFs controlling processes across time based on transcriptomic and epigenomic data. The workflow can also be used for processes other than differentiation such as drug treatments and disease progression. This study used EPIC-DREM for the first time in conjunction with ATAC-seq data. Previously, EPIC-DREM operated on ChIP-seq datasets (Gérard et al, 2018) and some of the tools integrated into this pipeline have been optimized for the use of other methods to detect accessible regions such as DNaseI-seq and NOMe-seq (Schmidt et al, 2017). EPIC-DREM can be applied to an ample range of epigenomic datasets that together with transcriptomic data can help to uncover key regulators and biological insights.

For the first time, we identified TFs controlled by SEs during mDAN differentiation. A total of 49 TFs were found to be under the control of SEs between day 30 and day 50 of differentiation (Dataset EV2). Of these, 17 were among the top TFs predicted by

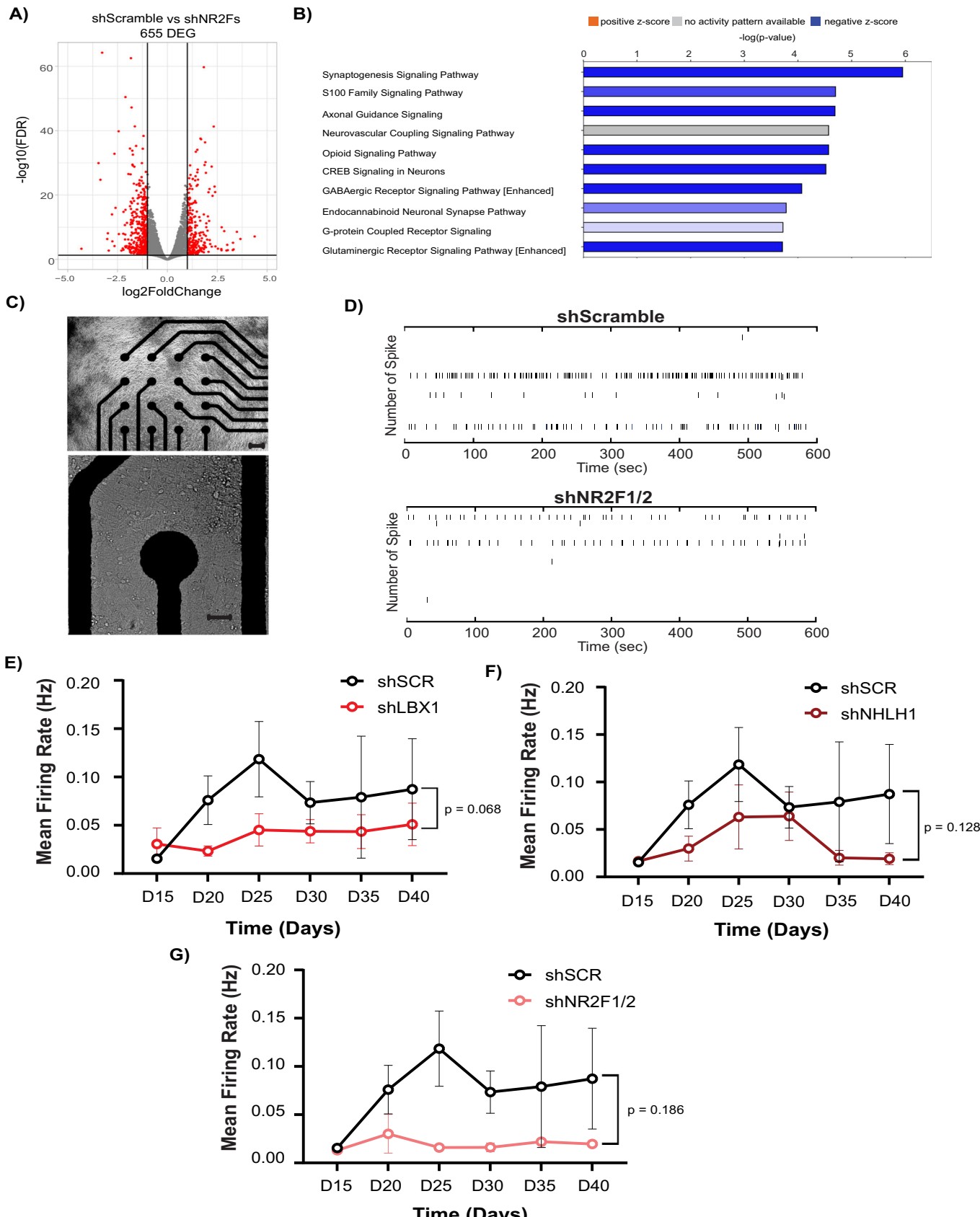

◄ **Figure 7. NR2F1/2 control synapse formation and neuronal activity.**

(A) Volcano plot showing DEGs from RNA-seq analysis of neurons at day 15 following a late transduction with shScramble or shNR2F1 and NR2F2. Black lines represent cut-off according to FDR <0.05 and absolute log$_2$-fold change >1. (B) Top 10 pathways from Ingenuity Pathway Analysis (IPA) based on significance of enrichment using DEGs from panel A. x-axis represents the −log10 p value of the enrichment. (C) Representative image of one 48-well MEA plate well (upper panel, scale bar = 100 µm) and a magnification of a single electrode surrounded by the neural network (lower panel, scale bar = 40 µm). (D) Raster plot of the neuronal spikes after 10 min of recording of shSCRAMBLE and shNR2F1/2. (E–G) Mean firing rate of the neurons following transduction with shSCRAMBLE in parallel with (E) shLBX1, (F) shNHLH1, or (G) shNR2F1/2 and measured every 5 days from day 15 to day 40. Data information: Data are representative of 3 (A, E–G) independent experiments. Error bars (E–G) correspond to ±1 standard deviation (SD) from the mean. Two-way ANOVA model was used for statistical analysis (E–G) with shSCRAMBLE was the reference sample. *p value <0.05, **p value <0.01, ***p value <0.001, and ns not significant. Source data are available online for this figure.

EPIC-DREM (Fig. 3B). Among the remaining 32 TFs that did not overlap with EPIC-DREM, we found additional promising candidates. For example, ZFHX4 and CSRNP3 are highly expressed in developing human mDANs in vivo (http://linnarssonlab.org/ventralmidbrain/; La Manno et al, 2016). These TFs were not selected in our analysis because their motifs are not known, namely, their TFBS in the accessible regions cannot be determined through footprinting, and the consequent prediction as regulators was not possible. This highlights a limitation of the approach used here, but simultaneously opens new research avenues for previously less studied TFs that seem to play an important role in neurogenesis.

Gene perturbation experiments were performed to validate the identified candidates: HOXB2, LBX1, NHLH1, NR2F1, and NR2F2. After TF depletion during differentiation, it was found that LBX1, NHLH1 and NR2F1/2 presented a stronger phenotype than a well-established and characterized TF involved in mDAN neurogenesis: LMX1B. Further characterization of the role of these TFs revealed that only LBX1 and NHLH1 inductions during differentiation were able to increase mDAN numbers, while induction time was critical for LBX1.

Overall, our results are in line with previous findings. Loss of function in the LBX1 protein, specifically in the protein–protein interaction domain, in addition to being lethal during development due to cardiac abnormalities in mice, has been shown to impair neurogenesis during the development of the neural tube (Decourtye et al, 2022). This finding can be correlated with what was observed during LBX1 KD in mDANs, where differentiating cells always struggle to survive. Moreover, the findings on the development of spinal cord interneurons, RTN, and dorsal neurons from the hindbrain have linked LBX1's function to cell fate decisions (Decourtye et al, 2022; Hernandez-Miranda et al, 2018; Gross et al, 2002). Here we have added a role for LBX1 in mDAN differentiation.

On the other hand, NHLH1 loss of function has been shown to be lethal in mice, but only in adult life. However, if accompanied by loss of NHLH2, mice die already at birth (Cogliati et al, 2002; Krüger et al, 2004). Furthermore, NHLH1 has been shown to control neurogenesis by regulating the expression of neuronal-specific genes during *Xenopus* development (Bao et al, 2000). Our results support an important role for NHLH1 in controlling mDAN differentiation, consistent with previous findings.

After seeing the relevance of LBX1, NHLH1, and NR2F1/2 during differentiation, transcriptomic analysis after TF depletion was performed to determine the main processes controlled by these TFs. LBX1 KD showed a clear downregulation of cholesterol biosynthesis-related genes together with SREBF1 and SREBF2. As SREBF1 is at the same time regulated by LXRα/β, and all of them have been previously related to mDAN neurogenesis, it was decided

to stimulate this pathway using GW3965. However, the treatment with this molecule was not sufficient to rescue the LBX1 KD effect or increase mDAN numbers in the differentiation protocol used for this study. It is possible that for GW3965 treatment to be effective, it should be either added from the beginning of differentiation or cells should be treated for a more extended period. We refrained from these experiments because the main effects of LBX1 KD were studied late in differentiation during the first 5 days of maturation.

On the other hand, as stimulation of TFs involved in cholesterol metabolism did not rescue the LBX1 KD effect, we focused on another enriched pathway, eIF signaling. This pathway is involved in translation, a process also downregulated by LBX1 KD. Exploration of the data for a common origin of cholesterol biosynthesis and translation downregulation revealed mTOR signaling inhibition as a possible explanation and in line with previous findings (Wang et al, 2011; Fonseca et al, 2014; Yecies et al, 2011; Liu and Sabatini, 2020; Wang et al, 2019). mTOR signaling was recently shown to play different roles in mDAN biology, controlling their morphology, dopamine release, and electrophysiology (Kosillo et al, 2022). Indeed, inhibiting mTOR signaling using rapamycin recapitulated most of the features observed by LBX1 KD.

Transcriptional profiling and more detailed analysis of neurons upon NHLH1 KD revealed a downregulation of the mature form of miR-124-3p. This relationship between NHLH1 and miR-124 has not been described before. miR-124 is a potent neurogenic microRNA that has been found to be downregulated in PD patients (Yang et al, 2021) and shown to be protective in PD animal models (Zhang et al, 2022; Saraiva et al, 2016). After determining the capacity of ABX464 in inducing miR-124 levels in neurons, the molecule was used to determine its effect during mDAN differentiation and as an attempt to rescue the NHLH1 KD. It was found that for inducing at least a 2-fold increase of miR-124-3p, the concentration needed was causing neurotoxicity, possibly due to the already high expression of miR-124 in the neuronal lineage. Independently of that, ABX464 treatment was able to increase mDAN numbers and to induce NHLH1 expression. Therefore, NHLH1 and miR-124 seem to be part of a positive feedback loop. To unveil potential applications of ABX464, its neurotoxicity should be reduced for example by chemical modification of the molecule. ABX464 is already in clinical trials for the treatment of rheumatoid arthritis and ulcerative colitis due to its anti-inflammatory potential (Vermeire et al, 2021; Daien et al, 2022). Namely, ABX464 safety and efficacy has been already evaluated and the repurposing of its use for PD could be taken into consideration to increase miR-124-3p levels as a neuroprotective treatment.

Finally, NR2F family TFs were found to contribute to regulation of neuronal activity in differentiating mDANs. This is consistent with previous findings on involvement of NR2F1 and NR2F2 in

development of different brain structures in various animal models, regulation of neurodevelopmental timing, and their association with neurodevelopmental disorders in humans (Yang et al, 2017; Bertacchi et al, 2019; Naka et al, 2008). Neither NR2F factor, nor NHL1 or LBX1, were found to change their expression in meta-analysis of PD post-mortem brain samples (Tranchevent et al, 2023). However, recent single nuclei RNA-seq analysis of human dopamine neuron subtypes from midbrains of PD patients and controls identified NR2F2 as a regulator of genes altered in neurons affected by PD (Kamath et al, 2022). Moreover, altered expression of NR2F2 has been observed in sporadic PD and some patient-derived cells lines, with elevated NRF2 expression increasing neurodegeneration in mouse model of PD (Kao et al, 2020). In addition to NR2Fs, also NHLH1 and LBX1 could play a role in diseases involving dopaminergic signaling. Indeed, polymorphisms at NHLH1 and LBX1 loci have been associated with schizophrenia and PD symptoms, respectively (Alfradique-Dunham et al, 2021; Trubetskoy et al, 2022). More detailed analysis will be required to elucidate the roles of these novel regulators of mDAN differentiation.

Importantly, it should be noted that none of the TFs investigated here in detail, namely NHLH1, LBX1, or NR2F1/2, are uniquely expressed in mDANs. Indeed, they show expression also in some other subtypes of neurons, and are likely to contribute to neurogenesis more broadly (Decourtye et al, 2022; Hernandez-Miranda et al, 2018; Gross et al, 2002; Yang et al, 2017; Bertacchi et al, 2019; Naka et al, 2008, Saunders et al, 2018). It will be critical to investigate both the general and cell type-specific roles of these factors for better understanding of their overall role in neurogenesis. Such analysis need to be performed both in vitro and in vivo, to ultimately develop improved strategies for the directed differentiation of both mDANs and other neuronal subtypes.

## Methods

### Cell lines

The human iPSC line GM17602 (Coriell) was used in this study as a control and for the generation of a tyrosine hydroxylase (TH) reporter cell line (TH-Rep1). This iPSC line was previously characterized in (Zanon et al, 2019) (called HFF) and used in (Rakovic et al, 2022) for the generation of the reporter line. Briefly, in the reporter line, the T2A coding sequence was fused with the mCherry open reading frame, and it was biallelically inserted in place of the stop codon in the endogenous TH locus using CRISPR/Cas9 editing (Fig. 1A). The second reporter iPSC line (TH-Rep2) was generated similarly using STBCi033-B cell line. In addition, for high-content imaging two non-reporter iPSC lines from healthy donors were used, namely K2531 (also known as LUEi013-A) and 17608/6 from NeuroBiobank Tübingen (Schöndorf et al, 2014). Cell lines were regularly tested for mycoplasma.

### Cell culture and differentiation

hiPSC maintenance, generation of small molecule neural precursor cells (smNPC) and differentiation towards mDANs are described in (Hanss et al, 2021). In Fig. 1A, a schematic representation of the mDAN differentiation protocol can be observed with the different media used and for how long the cells were kept in culture. This differentiation protocol is composed of three different media. smNPCs are incubated in Differentiation medium one containing 100 ng/ml of FGF8b, 1 μM of purmorphamine (PMA), and 200 μM of ascorbic acid which starts with smNPC and is used during the first eight days of differentiation. From day 8 until day 10 of differentiation cells are kept in Differentiation medium two, composed of 0.5 μM PMA and 200 μM of ascorbic acid. Lastly, maturation medium containing 200 μM of ascorbic acid, 10 ng/ml of brain-derived neurotrophic factor (BDNF), 10 ng/ml of glial cell-derived neurotrophic factor (GDNF), 500 μM of dcAMP, and 1 ng/ml of TGFβ3 which is used from day 10 until the desired time point. The molecules used in the three different media were mixed in N2B27 medium (1:1 of Dulbecco's modified Eagle medium/Nutrient Mixture F-12 [DMEM/F12] and Neurobasal medium supplemented with 1% Pen/Strep, 1% GlutaMAX, 1% B27 supplement minus vitamin A, and 0.5% N2 supplement, all from Gibco).

Astrocyte differentiation was induced based on the protocol from (Palm et al, 2015), with small changes. Briefly, smNPC were seeded in N2B27 medium complemented with 3 μM CHIR99021, 0.5 μM PMA and 150 μM ascorbic acid. After two days, fresh medium plus 20 ng/ml of FGF-2 was added to the smNPC culture. On day four, the cells were split into a new plate to start neural stem cell (NSC) generation. The medium used for the generation and maintenance of NSC contained DMEM/F-12, 1% Pen/Strep, 1% GlutaMAX, 1% B27 supplement serum-free (with vitamin A), 1% N2 supplement, 40 ng/ml EGF, 40 ng/ml FGF-2, and 1.5 ng/ul hLIF. The cells were kept in this medium for 3-4 passages. Then, astrocyte differentiation was started in DMEM/F-12 supplemented with 1% Pen/Strep, 1% GlutaMAX, and 1% FBS.

All the plates used were previously coated with Geltrex matrix from Gibco. Experiments where control cells failed to differentiate were excluded.

### Flow cytometry and FACS

On the day of analysis, medium was removed from the cells and Accutase (Gibco) was added to the well. Cells were incubated in Accutase at 37 °C until detachment (10–30 min). Then, 2 volumes of DMEM/F-12 were added to the well. Cells were gently pipetted up and down until dissociation and collected in a 15 ml Falcon tube after passing them through a 50 μm cell strainer to obtain a single-cell suspension. Falcon tubes were centrifuged 3 min at $300 \times g$ and room temperature (RT). Pellets were resuspended in PBS and cells were transferred to a 1.5 ml Eppendorf tube. 4',6-diamidino-2-phenylindole (DAPI) was added to the cells at 5 μg/ml and incubated for 5 min at 4 °C in a tube rotator. After incubation, the cells were washed twice with 2% (w/v) BSA in PBS. Then, cells were ready for either flow cytometry analysis or fluorescence-assisted cell sorting (FACS). For flow cytometry analysis, the BD LSRFortessa™ cell analyzer was used. For FACS, the BD FACSAria™ III sorter was used. FlowJo version 10 software was used for data processing and generation of plots.

### Total RNA extraction, cDNA synthesis and RT-qPCR

For total RNA extraction, Quick-RNA miniprep kit from Zymo Research was used as per manufacturer's instructions. When total

RNA had to be extracted from FACS sorted cells, Quick-RNA microprep kit from Zymo Research was used. Briefly, to avoid RNA degradation due to FACS, cells were collected in batches for no longer than 10 min. Then, sorted cells were pelleted by centrifugation for 3 min at $500 \times g$ and 4 °C. Lysis buffer from the kit was added to the pellet until 150,000–300,000 cells were collected. RNA extraction was finalized following the manufacturer's instruction.

For cDNA synthesis, amounts from 200 ng to 1 µg of total RNA were used, depending on sample availability. To perform the reaction, dNTPs (0.5 mM, ThermoFisher), oligo dT-primer (2.5 µM), 1 µl RevertAid reverse transcriptase (200 U/µl, Thermo-Fisher), and 1 µl Ribolock RNase inhibitor (40 U/µl, Thermo-Fisher) were mixed with the proper amount of total RNA in a final volume of 40 µl. cDNA synthesis was performed at 42 °C for 1 h. Reaction was terminated by incubating the reaction at 70 °C for 10 min. cDNA was diluted 1:10, 1:5 or 1:2 in DNase/RNase free water, depending on the amount of total RNA used (1 µg, 500 ng, or 200 ng, respectively). Diluted cDNA was stored at −20 °C.

RT-qPCR was performed in an Applied Biosystems 7500 Fast Real-Time PCR system. For the reaction, 5 µl of diluted cDNA was mixed with 1×Absolute Blue qPCR SYBR green low ROX mix (ThermoFisher) and 500 nM primer concentration, in a final volume of 20 µl per well using AmpliStart 96-well plate (Westburg). PCR reaction had the following settings: 95 °C for 15 min and then 40 cycles of 95 °C for 15 s, 55 °C for 15 s and 72 °C for 30 s. The $2^{-(\Delta\Delta Ct)}$ method was used to calculate gene expression levels. $\Delta\Delta Ct$ was calculated using the following formula: $(\Delta Ct_{(target\ gene)} - \Delta Ct_{(housekeeping\ gene)}) - (\Delta Ct_{(target\ gene)} - \Delta Ct_{(housekeeping\ gene)})_{reference\ condition}$. ACTB was used as the housekeeping gene. GraphPad Prism 9 was used to create plots and perform statistics. Data was always normalized to the reference condition and one sample $t$ test was performed to determine significance.

## Omni-ATAC

The Omni-ATAC protocol was performed with 50,000 cells (or 25,000 cells for day 50 neurons) with minor modifications from (Corces et al, 2017). Using TDE1 Tagment DNA Enzyme and TD Buffer from Illumina. Cells were either derived from FACS or directly from cell culture plates after detachment with Accutase. After amplification with primer sets as described in (Buenrostro et al, 2013), libraries underwent size selection using AMPure XP beads (Beckman Coulter) to remove large fragments. Libraries were stored at −20 °C. For library quality control, the Agilent High Sensitivity DNA kit was used in a 2100 Bioanalyzer instrument. Libraries used for sequencing presented a fragment distribution starting from ~200 bp until around 1000 bp, with nucleosomal pattern peaks.

## Low-input ChIP-seq and calling of SEs

To perform low-input chromatin immunoprecipitation (ChIP) for H3K27ac, the Low Cell ChIP-Seq Kit from Active Motif was used (53084). For smNPC and sorted neurons, a total of 150,000 and 200,000 cells were used, respectively. The protocol was performed according to manufacturer's instructions, with minor changes. Sonication of the cells was performed using a Bioruptor® Pico sonication device from Diagenode. The settings used for sonication

were 40 cycles of 30 s off and 30 s on at 8 °C. After sonication, 20% of the sonicated sample volume was saved as input. For the immunoprecipitation (IP) reaction, a total of 4 µg of H3K27ac antibody was used per reaction. Details about the antibody can be found in the Antibodies section. Input samples that were collected after sonication were processed together with IP reactions starting from the step where reversal of cross-links and DNA purification was performed. After this point, samples were processed in parallel. Libraries were stored at −20 °C until quality control and sequencing. For library quality control, the same procedure as in the Omni-ATAC protocol was used. For low-input ChIP, good libraries presented an average of 600 bp fragment distribution. The low-input ChIP-seq protocol was validated by comparing the identified SEs to those previously detected in smNPC via regular high-input ChIP-seq methods (Walter et al, 2021). SEs were considered as regions larger than 10 kb.

## Immunocytochemistry

PhenoPlate 96-well (PerkinElmer) previously coated with geltrex was used for immunocytochemistry. Cells were fixed in 4% paraformaldehyde for 15 min at RT. Then, cells were washed three times with PBS. For permeabilization, cells were incubated 1 h at RT in PBS, 0.4% Triton-X, 10% goat serum and 2% BSA. After 1 h, cells were washed twice with PBS. Primary antibody was diluted in PBS, 0.1% Triton-X, 1% goat serum and 0.2% BSA. Cells were incubated with primary antibody shaking overnight at 4 °C. Next day, cells were washed three times with PBS. Then, secondary antibody was diluted in the same buffer as the primary and incubated for 2–3 h at RT. Finally, cells were washed three times with PBS. In the first wash, DAPI was added to the PBS (5 µg/ml) and incubated for 15 min at RT. Images were taken using a Zeiss spinning disk confocal microscope. Image processing was done in ImageJ.

## High-content imaging

For each condition, at day 11, 120,000 transduced cells were seeded in duplicate on 96-well format (PerkinElmer, 6055308). At day 14, cells were fixed and stained as described above, using rabbit polyclonal anti-TH 1:250 (Merck-AB152), chicken polyclonal anti-MAP2 1:1000 (Abcam-ab5392) with their respective secondary antibodies goat anti-Rabbit IgG Alexa Fluor™ 647 (Invitrogen—A27040) and goat anti-Chicken Alexa Fluor™ 568 (Invitrogen—A11041) 1:1000 and DAPI nuclear staining.

Twelve image $z$-stacks per well were taken at ×20 magnification using the CellVoyager CV8000 High-Content Screening System (Yokogawa). The $z$-stacks include three planes separated by 3.2 µm steps. DAPI was excited with a 405 nm laser and detected behind a 445/45 bandpass filter. Alexa568 was excited with a 561 nm laser and detected behind a 600/37 bandpass filter. Alexa647 was excited with a 640 nm laser and detected behind a 676/29 bandpass filter.

Using an in-house Matlab analysis pipeline, signal masks for both TH and MAP2 were generated after thresholding. Area with signal above threshold was quantified for TH, MAP2, and DAPI signal. TH area was normalized to neuron numbers using the ratio of TH area over MAP2 area. The analysis code is available upon request.

## Bacterial culture, plasmid extraction and lentivirus production

Glycerol stocks of bacteria containing the plasmid of interest were taken from −80 °C. Without letting the glycerol stocks thaw and with the help of a P10 pipette tip, bacteria were added to a polypropylene graduated culture tube (Roth) containing 5 ml of LB Broth medium (20.6 g/l, Roth) supplemented with Ampicillin (100 μg/ml). Bacteria were incubated at 37 °C and 120 rpm shaking for 5 h. Then, bacteria were transferred to an Erlenmeyer containing 150 ml of LB Broth medium supplemented with ampicillin. Erlenmeyer was incubated overnight at 37 °C and shaken at 120 rpm. After bacterial expansion, plasmid extraction was performed using NucleoBond Xtra Midi EF as per manufacturer's instructions.

For lentivirus production, 8 million HEK293T cells were seeded in a T75 flask using 15 ml of DMEM (Gibco) supplemented with 1% Pen/Strep, and 10% FBS and transfected the next day for lentiviral production. Third-generation lentiviral particles were produced. Briefly, 4 μg of pMDG, 2 μg of pMDL, 2 μg of pREV, and 8 μg of the plasmid of interest were mixed with 200 μl of $CaCl_2$ (1 M, Sigma). Volume was completed with sterile water up to 800 μl. The 800 μl of the plasmid mixture was mixed with 800 μl of HEPES buffered saline (Sigma) by making bubbles slowly in a dropwise manner. This transfection mixture was incubated for 20 min at RT. In the meanwhile, 16 μl of 25 mM chloroquine was added to the T75 flask containing the HEK293T cells and incubated for a minimum of 5 min to facilitate transfection. Next, the transfection mixture was added to the HEK293T cells. After 4–6 h, medium was removed from HEK293T cells and 14 ml of fresh medium was added. After 48 h, lentiviral particles were ready for collection. HEK293T medium from the flask was collected in a 15 ml Falcon tube and centrifuged for 10 min at 2000 rpm and 4 °C. The supernatant was cleared by filtering through a 0.45 μm filter (Sartorious). Filtered lentiviral particles were aliquoted in cryovials (1 ml aliquots) and stored at −80 °C.

## Transduction

Two different approaches for transduction of differentiating neurons were used in this study: early and late transductions. All the lentiviral particles used contained a GFP reporter that helped control for transduction efficiency. For overexpression constructs a codon-optimized cDNA sequence of the TF was used. Lentiviral particles were previously tested to adjust transduction efficiencies to ~80%, determined by GFP positive cells using flow cytometry.

For early transduction, smNPC were seeded in a 6-well plate with a density of 1–2 million cells per well and differentiation was started by seeding them directly in a differentiation medium. Next day, medium was removed and lentiviral particles were added to the cells in a final volume of 1 ml. Plate was sealed with parafilm and centrifuged for 10 min at $250 \times g$ and RT for spinfection. After spinfection, 1 ml of differentiation medium was added to the cells. Lentiviral particles were incubated overnight. Next day, lentiviral particles were removed, and fresh differentiation medium was added to the cells. Differentiation continued until the day of analysis.

For late transduction, differentiating cells were split on day 8 of differentiation into a 6-well plate at a density of 3 million cells per

well. The next day, the medium was removed, and transduction was performed as described for early transduction. Differentiation continued until the day of analysis. Experiments where cells were not successfully transduced were excluded.

## GW3965, ABX464, and Rapamycin treatments

GW3965-HCl (Selleckchem), ABX464 (Selleckchem), and Rapamycin (Selleckchem) molecules were tested to determine their working concentration in our cultures. For GW3965-HCl, 0.1 μM was sufficient to induce SREBF1 mRNA levels. For ABX464, 10 μM was sufficient to induce miR-124-3p. For Rapamycin, 10 nM was sufficient to downregulate SREBF1 mRNA levels.

The effect of the molecules was tested during normal differentiation and under LBX1 or NHLH1 knock-down (KD) conditions. Briefly, neuronal differentiations were started and on day 8 of differentiation, cells were split into a 6-well plate with a density of 3 million cells per well. Next day, late transduction with shLBX1, shNHLH1 or shScramble was performed as described above. On day 10 of differentiation, transduced cultures were treated either with GW3965, ABX464, or rapamycin. Cells transduced with shLBX1 were treated with GW3965 or Rapamycin, while cells transduced with shNHLH1 were treated with ABX464. Treatments continued until day 15, when cells were analyzed.

## TaqMan assay

TaqMan assay was performed to determine miR-124-3p levels using TaqMan™ MicroRNA Reverse Transcription Kit (Thermo-Fisher), TaqMan™ MicroRNA Assay hsa-miR-124-3p (Thermo-Fisher, AssayID 003188_mat), TaqMan™ MicroRNA Control Assay U6 snRNA (ThermoFisher, AssayID 001973), TaqMan™ MicroRNA Assay hsa-miR-423-5p (ThermoFisher, AssayID 002340), and TaqMan™ Fast Advanced Master Mix (ThermoFisher). First, reverse transcription reaction was performed using 0.15 μl of 100 mM dNTPs, 1 μl of MultiScribe™ Reverse Transcriptase (50 U/μl), 1.5 μl of 10× Reverse Transcription buffer, 0.19 μl of RNase inhibitor (20 U/μl), 10 ng of total RNA, and 3 μl of RT primer (either U6 or miR-124-3p primer) in a final volume of 15 μl. Reactions were incubated for 30 min at 16 °C and then 30 min at 42 °C. Reaction was terminated by incubating the tubes for 5 min at 85 °C. Then, PCR was performed using the same plates and machine as for RT-qPCR. For the PCR reaction, 1 μl of 20× TaqMan MicroRNA Assay (either from U6 or miR-124-3p), 1.33 μl from RT reaction, and 10 μl of TaqMan™ Fast Advanced Master Mix were mixed in a final volume of 20 μl. The PCR reaction settings were: 10 min at 95 °C followed by 40 cycles of 15 s at 95 °C and 1 min at 60 °C. To calculate miR-124-3p levels, the $2^{-(\Delta\Delta Ct)}$ method was used again, where U6 snRNA represented the housekeeping gene. GraphPad Prism 9 was used as described for RT-qPCR analysis.

## Sequencing

Prior RNA-seq, RNA quality was determined by using the Agilent RNA 6000 Nano kit in an Agilent 2100 Bioanalyzer machine. Samples selected for sequencing had a RIN value >7. RNA-seq from time course data, including smNPC, mDANs and astrocytes

samples, was done using the TruSeq Stranded mRNA library prep kit, single-end 75 bp read length, and a NextSeq500 machine.

RNA-seq from KD samples was done using an Illumina stranded mRNA library prep. Kit, paired-end 50 bp read length, and a NovaSeq6000 machine.

ATAC-seq samples were sequenced on a NextSeq500 machine using paired-end 75 bp read length.

ChIP-seq samples were sequenced on a NextSeq500 machine using single-end 75 bp read length.

## RNA-seq analysis

For the time course RNA-seq, including smNPC, mDANs at days 15, 30 and 50 of differentiation, non-mDAN at days 15 and 50 of differentiation, and astrocytes at day 65 of differentiation the following tools were used. Raw fastq files were assessed for quality using FastQC (https://www.bioinformatics.babraham.ac.uk/projects/fastqc/). Summary of sample quality controls was obtained using MultiQC (Ewels et al, 2016). Next, the Paleomix pipeline was used for trimming sequencing adapters using AdapterRemoval (Schubert et al, 2016, 2014). After adapter removal, ribosomal RNA was filtered from the data using SortMeRNA (Kopylova et al, 2012). Alignment to the reference genome was done using STAR (Dobin et al, 2013). BAM files were validated using Picard (https://broadinstitute.github.io/picard/). Quality reads (≥Q30) were filtered using SAMtools (Danecek et al, 2021). Gene counts were obtained using FeatureCounts from the Rsubread package (Liao et al, 2019). Differential expression analysis was done using the R package DESeq2 (Love et al, 2014). For more information about the specific versions and settings used for the different tools, please refer to our repository (RNA-seq folder, RNA-seq_DataAnalysis_TimeCourse.rmd script). Genome version and annotation were GRCh38 patch 12 and Gencode human release 31, respectively.

For the RNA-seq data from the different TF knock-down experiments, including LBX1 KD, NHLH1 KD and NR2F1/2 KD, a snakemake pipeline was used (Köster et al, 2021). This pipeline includes the tools STAR, SAMtools, FastQC, FastQ Screen, AdapterRemoval, Rsubread, DESeq2, ggplot2 and apeglm (Wickham, 2016; Wingett and Andrews, 2018; Zhu et al, 2019). For more details about the pipeline, please refer to our repository (RNA-seq folder, RNA-seq_DataAnalysis_TF_KDs.rmd script). Genome version and annotation were GRCh38 release 102.

## ATAC-seq analysis

For the time course ATAC-seq, including smNPC, mDANs on days 15, 30, and 50 of differentiation, non-mDAN on days 15 and 50 of differentiation, and astrocytes on day 65 of differentiation the following tools were used. Raw fastq files were assessed for quality using FastQC. A summary of sample quality control was obtained using MultiQC. Using the Paleomix pipeline, trimming of sequencing adapters was done using AdapterRemoval and mapping to the reference genome with BWA (Li and Durbin, 2009). BAM files were validated using Picard. Quality reads (≥Q30) were filtered using SAMtools. Peak calling was performed using Genrich (https://github.com/jsh58/Genrich). For more information about the specific versions and settings used for the different tools, please

refer to our repository (README.md file inside the ATAC-seq folder). Genome version was GRCh38 patch 1.

## ChIP-seq analysis

For H3K27ac ChIP-seq, including smNPC, mDANs at days 30 and 50, and non-mDANs at day 50, the following tools were used. After merging R1 and R2 raw fastq files as described by Active Motif, the new fastq files were assessed for quality using FastQC. Using the Paleomix pipeline, trimming of sequencing adapters was done using AdapterRemoval and mapping to the reference genome with BWA. BAM files were validated using Picard. Quality reads (≥Q30) were filtered using SAMtools. New BAM files were sorted according to mapping position using SAMtools prior the molecular identifier de-duping step. For de-duping, a perl script provided by Active Motif was used (not provided, it should be requested from the manufacturer. Script name rmDupByMids.pl.txt, version from 2019). For calling enhancers and super-enhancers HOMER was used (Heinz et al, 2010). For more information about the specific versions and settings used for the different tools, please refer to our repository (ChIP-seq folder, Lowinput_ChIP-seq_analysis.rmd script). Genome version was GRCh38 patch 12.

## EPIC-DREM analysis

EPIC-DREM was applied as a snakemake pipeline. As an input, pre-processed BAM files from the ATAC-seq analysis and the gene counts from the previously described RNA-seq analysis were used, including smNPC, and mDANs at days 15, 30 and 50. ATAC-seq peak calling was performed with Genrich over replicates. The Regulatory Genomics Toolbox was used to identify footprints in called peaks (Li et al, 2019) and subsequently, TF-gene affinities were calculated using TEPIC (Schmidt et al, 2017). The resulting time-point-specific lists of TF–gene links were merged and filtered according to expression, removing links from unexpressed TF. TF was considered unexpressed if it presented with a transcripts per million (TPM) value <1 in all analyzed time points. Time-point-specific GRN were identified with interactive Dynamics Regulatory Events Miner (iDREM) (Ding et al, 2018). Results from iDREM were further processed in R for visualization and GO enrichment was performed using clusterProfiler (Wu et al, 2021). For more details about the different parameters used, please refer to our repository (EPIC-DREM folder).

## Multielectrode array (MEA) recordings

Neuronal network activity was recorded using a multi-well MEA system (Maestro, Axion BioSystems, Atlanta, GA). The MEA plates were composed of 48 wells, each containing a square grid of 16 PEDOT electrodes (50 μm electrode diameter; 350 μm center-to-center spacing) of recording area. Spiking activity from networks grown onto MEAs was recorded and monitored using Axion BioSystems hardware (Maestro1 amplifier and Middle-man data acquisition interface) and the Axion's Integrated Studio software in Spontaneous Neural Configuration (AxIS 2.1). Raw data were digitized and stored on a hard disk for subsequent offline analysis.

The day before culture preparation, MEAs were primary coated by depositing a 50 μl drop of Poly(Ethyleneimine) (PEI) (0,07% in

1× borate buffer, Sigma-Aldrich) over each recording area and subsequently incubated for 1 h. After three washes with sterile deionized water, MEAs were secondary coated by depositing a 50 μl a drop of Geltrex (1:100 diluted in culture medium, ThermoFisher Scientific) over each recording area and subsequently incubated for 1 h, before the cell seeding. Experiments were performed in culture medium maintained at 37 °C.

At day 15, and there on every 5 days up to day 40 as indicated, MEA plates were set on the Maestro apparatus and their spontaneous activity recorded for 10 min in a standard culture medium (basal condition) after 5 min of activity stabilization. Spikes were computed using the Axion BioSystems software NeuralMetricTool. Only the wells containing ≥1 active electrode (≥5 spikes/min) were retained to study firing and bursting properties for further analysis. Electrodes that recorded less than five spikes/min were deemed inactive and excluded for the analysis. Data are given as means ± SEM for $n$ = sample size. Condition groups were compared by using two-way ANOVA, followed by Tukey's multiple comparison tests. Statistical analysis was carried out by using the Prism software (GraphPad Software, Inc.).

## Primers

| Primer | sequence (5′ → 3′) |
|---|---|
| hACTB_F | AAACTGGAACGGTGAAGGTG |
| hACTB_R | AGAGAAGTGGGGTGGCTTTT |
| hNR2F1_F | GAGCAGGTGGAGAAGCTCAA |
| hNR2F1_R | CAGGCGTCTGACGTGAACAG |
| hNR2F2_F | AGGCGCTGCACGTTGAC |
| hNR2F2_R | AGGCATCTGAGGTGAACAGGACTA |
| hNHLH1_F | ACGCTACCCCTGAGAGTCTAGAAA |
| hNHLH1_R | TCTGGGTGCTCAAGGCTCAT |
| hLBX1_F | AAGGCCGCGACGGTATG |
| hLBX1_R | GCGACTTTCGCCGCTTCTTA |
| hSOX4_F | CCTAATTTCTCCATGTTTACACTTCAAT |
| hSOX4_R | GTGGACACTGGTGGCAGGTT |
| hHOXB2_F | CCGAGGAAGAGCTGGATTTTT |
| hHOXB2_R | GTTAGGGAAACTGCAGGTCGAT |
| hLMX1B_F | GCCGAAAGGTCCGAGAGA |
| hLMX1B_R | CTTCTTCATCTTTGCTCTTTGGTT |
| hNHLH1_OE_F | CCGACAAGAAGCTCTCCAAGA |
| hNHLH1_OE_R | TGGTTCAGGTAGGAGATATAGCAGATG |
| hSREBF1_F | GCTCCTCCATCAATGACAAAATC |
| hSREBF1_R | TGCAGAAAGCGAATGTAGTCGAT |
| hSREBF2_F | CCTGTCATTCGAGTCAGGTTCTG |
| hSREBF2_R | CAATCACACCATTTACCAGCCATA |
| hSCARB2_F | CTACAGGGAACTCAGAAACAAAGCA |
| hSCARB2_R | CCAACAGATTGGTCTCGTTCAA |
| hGAPDH_F | GCATCCTGGGCTACACTGAG |
| hGAPDH_R | GGTGGTCCAGGGGTCTTACT |

## Antibodies

| Antigen | Species | Dilution | Cat No. |
|---|---|---|---|
| TH | Rabbit | 1:250 | Sigma T8700-1VL |
| NeuN Alexa fluor 488 conjugated | Mouse | 1:250 | Sigma MAB377X |
| Histone H3 acetyl K27 (H3K27ac) | Rabbit | ** | Abcam Ab4729 |
| TH (Yokogawa) | Rabbit | 1:250 | Merck-AB152 |
| MAP2 | Chicken | 1:1000 | Abcam-ab5392 |
| Rabbit IgG Alexa Fluor 647 | Goat | 1:1000 | Invitrogen—A27040 |
| Chicken IgG Alexa Fluor 568 | Goat | 1:1000 | Invitrogen—A11041 |

## Bacterial glycerol stocks

All bacterial glycerol stocks are from GeneCopoeia. The different shRNA constructs used were:

| Name | Vector | shRNA sequence | Cat No |
|---|---|---|---|
| Scramble shRNA (shScramble) | psi-LVRU6GP | GCTTCGCGCC GTAGTCTTA | CSHCTR001-LVRU6GP |
| shRNA targeting HOXB2 (shHOXB2) | psi-LVRU6GP | GGTATTACTGA ATTAGCGTTT | HSH008988-LVRU6GP-b |
| shRNA targeting LMX1B (shLMX1B) | psi-LVRU6GP | GGGTGACTACG AGAAGGAGAA | CS-HSH101934-LVRU6GP-01-c |
| shRNA targeting NHLH1 (shNHLH1) | psi-LVRU6GP | GCTATATCTCC TACCTGAACC | HSH011829-LVRU6GP-a |
| shRNA targeting NR2F2 (shNR2F2) | psi-LVRU6GP | GGAGGAACCAC ATATAACACT | CS-HSH110558-LVRU6GP-01-c |
| shRNA targeting NR2F1 (shNR2F1) | psi-LVRU6GP | CCGCAGGAACT TAACTTACAC | HSH110557-LVRU6GP-b |
| shRNA targeting LBX1 (shLBX1) | psi-LVRU6GP | GACGTAGAGTC CGCCAAGAAA | HSH000806-LVRU6GP-d |

The different overexpression constructs used were:

| Name | Vector | Accession No. | Cat No |
|---|---|---|---|
| Control empty vector (GFP OE) | pEZ-Lv228 | ** | EX-NEG-Lv228 |
| NHLH1 overexpression (NHLH1 OE) | pEZ-Lv228 | NM_005598 | EX-F0569-Lv228 |
| LBX1 overexpression (LBX1 OE) | pEZ-Lv228 | NM_006562 | EX-T1288-Lv228 |
| NR2F2 overexpression (NR2F2 OE) | pEZ-Lv228 | NM_021005 | EX-C0221-Lv228 |

## Data availability

The source RNA-seq, ATAC-seq and ChIP-seq fastq files have been deposited at https://ega-archive.org/, under the accession number EGAD00001009288. Additional intermediate files such as RNA-seq counts, ATAC-seq peaks and bigwig files, and ChIP-seq H3K27ac SEs and bigwig files, can be provided upon request. The images from the high-content imaging analysis have been deposited in the BioStudies repository with the accession number S-BIAD900.

Analysis code can be found in the following repository: https://gitlab.lcsb.uni.lu/borja.gomezramos/gomezramos_et_al_2023/-/tree/main/. Data used in the preparation of this article were obtained on May, 22, 2023 from the Parkinson's Progression Markers Initiative (PPMI) database (www.ppmi-info.org/access-data-specimens/download-data), RRID: SCR_006431. For up-to-date information on the study, visit www.ppmi-info.org.

## Peer review information

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

## Acknowledgements

The computational analysis presented in this paper were carried out using the HPC facilities of the University of Luxembourg. We thank the Bioimaging Platform of the Luxembourg Centre for Systems Biomedicine for their support. All schematic representations were created with BioRender.com. This work was supported by the Luxembourg National Research Fund within the National Centre of Excellence in Research on Parkinson's Disease (NCER-PD; FNR/ NCER13/BM/11264123) and the PEARL program (FNR/P13/6682797 to RK). BGR and NdL were funded by the Luxembourg National Research Fund through the PARK-QC doctoral training unit (PRIDE17/12244779/PARK-QC); LS, BGR, and JO have received funding from Fondation du Pélican de Mie et Pierre Hippert-Faber and Luxembourg Rotary Foundation. The genome-editing platform in Lübeck is supported by the DFG (FOR 2488 to AR). DF and SB were supported by Luxembourg National Research Fund (CORE C21/DM/ 15839823/Astraging). GA is supported by the Luxembourg National Research Fund CORE grant C21/BM/15850547/PINK1-DiaPDs. PPMI—a public–private partnership—is funded by the Michael J. Fox Foundation for Parkinson's Research and funding partners, including 4D Pharma, Abbvie, AcureX, Allergan, Amathus Therapeutics, Aligning Science Across Parkinson's, AskBio, Avid Radiopharmaceuticals, BIAL, Biogen, Biohaven, BioLegend, BlueRock Therapeutics, Bristol-Myers Squibb, Calico Labs, Celgene, Cerevel Therapeutics, Coave Therapeutics, DaCapo Brainscience, Denali, Edmond J. Safra Foundation, Eli Lilly, Gain Therapeutics, GE HealthCare, Genentech, GSK, Golub Capital, Handl Therapeutics, Insitro, Janssen Neuroscience, Lundbeck, Merck, Meso Scale Discovery, Mission Therapeutics, Neurocrine Biosciences, Pfizer, Piramal, Prevail Therapeutics, Roche, Sanofi, Servier, Sun Pharma Advanced Research Company, Takeda, Teva, UCB, Vanqua Bio, Verily, Voyager Therapeutics, the Weston Family Foundation and Yumanity Therapeutics.

## Author contributions

**Borja Gomez Ramos**: Conceptualization; Formal analysis; Investigation; Visualization; Methodology; Writing—original draft; Writing—review and editing. **Jochen Ohnmacht**: Conceptualization; Formal analysis; Supervision; Funding acquisition; Investigation; Visualization; Methodology; Writing—review and editing. **Nikola de Lange**: Data curation; Software; Formal analysis; Investigation; Visualization; Methodology; Writing—review and editing. **Elena Valceschini**: Formal analysis; Validation; Investigation; Visualization; Methodology; Writing—review and editing. **Aurélien Ginolhac**: Data curation; Software; Formal analysis; Investigation; Visualization; Methodology; Writing—review and editing. **Marie Catillon**: Formal analysis; Investigation; Visualization; Methodology; Writing—review and editing. **Daniele Ferrante**: Formal analysis; Investigation; Visualization; Methodology; Writing—review and editing. **Aleksandar Rakovic**: Resources; Methodology; Writing—review and editing. **Rashi Halder**: Formal analysis; Methodology; Writing—review and editing. **Francois Massart**: Formal analysis; Investigation; Methodology; Writing—review and editing. **Giuseppe Arena**: Formal analysis; Investigation; Methodology; Writing—review and editing. **Paul Antony**: Software; Formal analysis; Methodology; Writing—review and editing. **Silvia Bolognin**: Resources; Supervision; Writing—review and editing. **Christine Klein**: Resources; Supervision; Funding acquisition; Methodology; Writing—review and editing. **Roland Krause**: Resources; Software; Formal analysis; Supervision; Methodology; Writing—review and editing. **Marcel H Schulz**: Conceptualization; Software; Formal analysis; Supervision; Investigation; Visualization; Writing—review and editing. **Thomas Sauter**: Conceptualization; Resources; Supervision; Funding acquisition; Writing—review and editing. **Rejko Krüger**: Conceptualization; Resources; Supervision; Funding acquisition; Project administration; Writing—review and editing. **Lasse Sinkkonen**: Conceptualization; Resources; Formal analysis; Supervision; Funding acquisition; Investigation; Visualization; Writing—original draft; Project administration; Writing—review and editing.

## Disclosure and competing interests statement

The authors declare no competing interests.

# Expanded View Figures

## A)

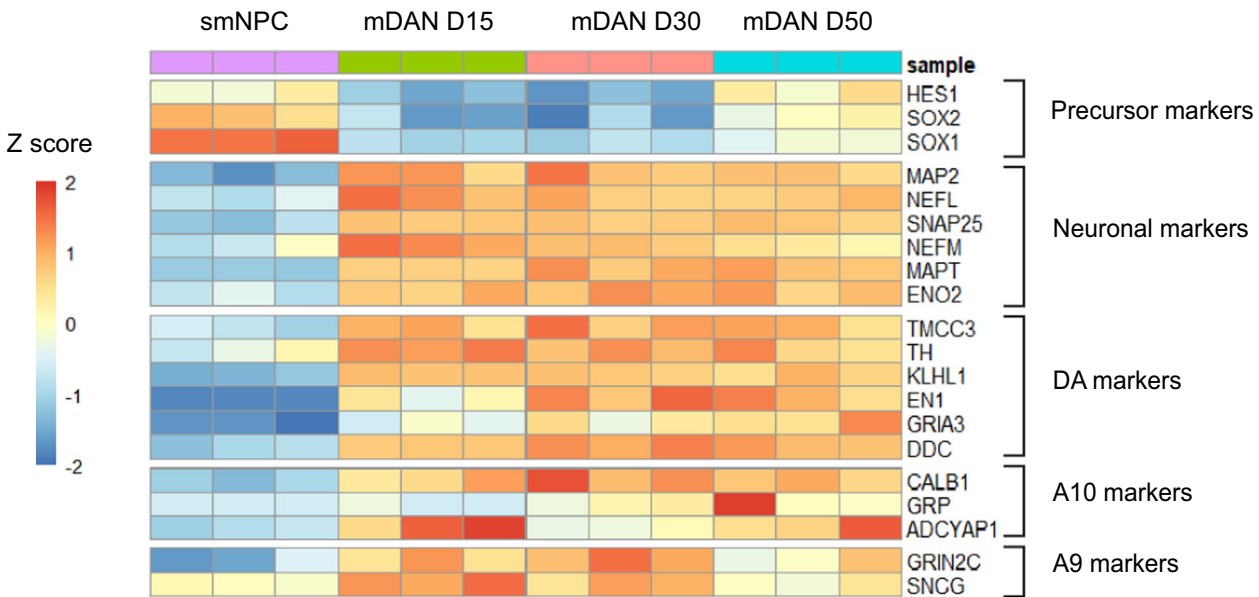

## B)

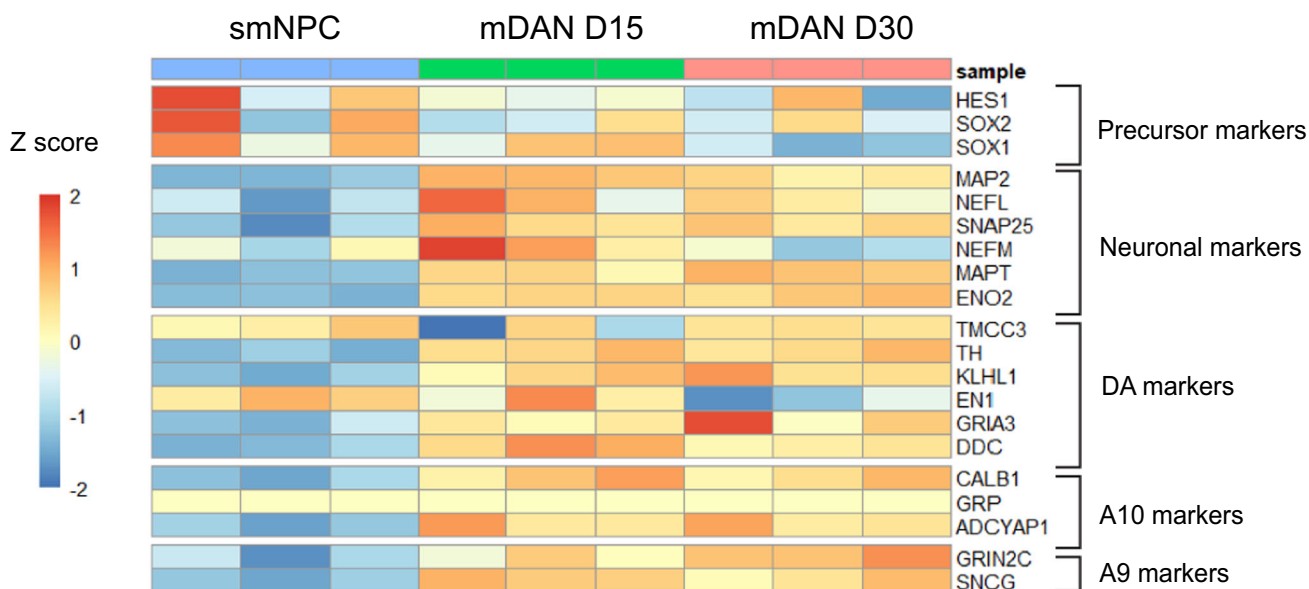

**Figure EV1.** **Expression of cell type-specific marker genes during mDAN differentiation.**

(A) Heatmap showing the expression of cell-type-specific markers of smNPCs, neurons, mDANs, and mDAN subtypes selected from literature (Anderegg et al, 2015; La Manno et al, 2016) in TH-Rep1 cell line. (B) Heatmap showing the expression of cell-type-specific markers from panel A in TH-Rep2 cell line. GRP A10 marker gene was not expressed in this iPSC cell line.

A)

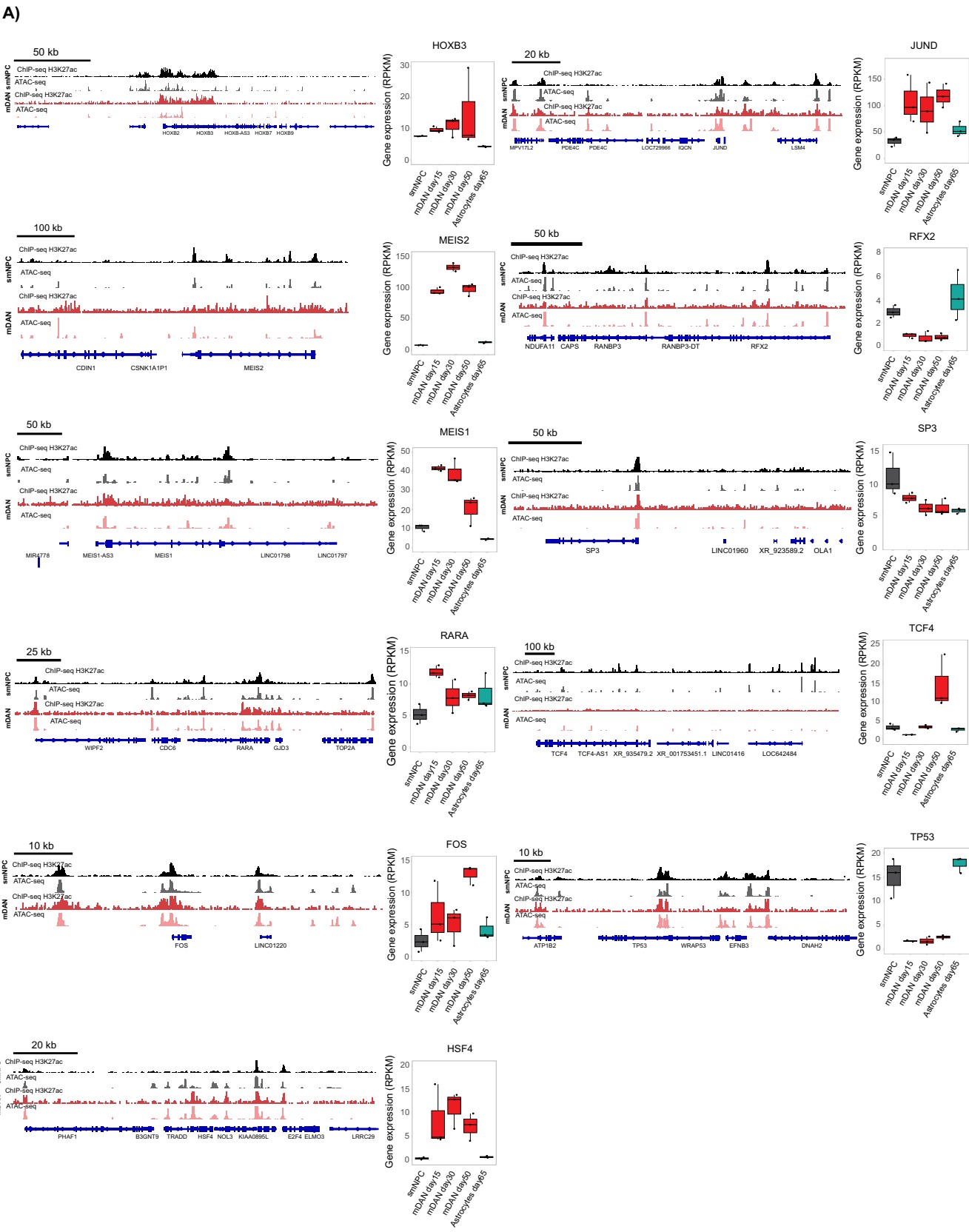

◀ **Figure EV2. TFs controlled by super-enhancers in mDANs.**

H3K27ac signal and chromatin accessibility profiles together with the expression dynamics during mDAN differentiation of the 11 additional TFs under the control of SEs between day 30 and day 50 and predicted by EPIC-DREM. ATAC-seq and ChIP-seq tracks are plotted under the same scale per dataset for comparison purposes. Data are representative of 3 independent experiments. Boxplots illustrate the distribution of data as described in Fig. 1.

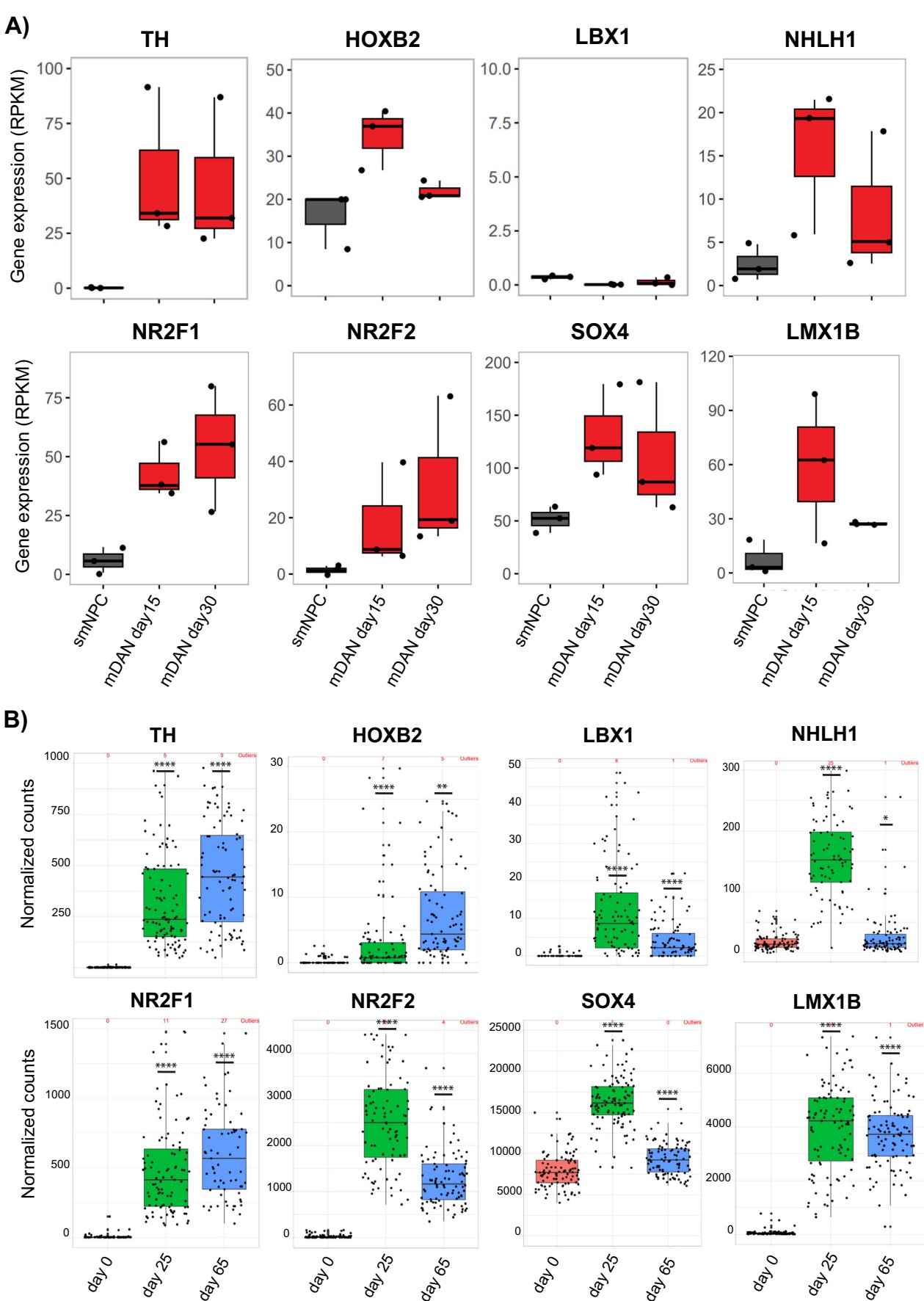

**Figure EV3.  Expression dynamics of candidate TFs in mDAN differentiation across different cell lines.**

(A) Expression dynamics of the novel candidate TFs during mDAN differentiation of the TH-Rep2 cell line. (B) Expression dynamics of the novel candidate TFs in 95 independent iPSC-lines during mDANs differentiation. The RNA-seq data was kindly provided by the Foundational Data Initiative for Parkinson's Disease (FOUNDIN-PD) consortium (Bressan et al, 2023). Data information: Data are representative of 3 (A) and 95 (B) independent experiments. Two samples *t* test was used for statistical analysis (B) with day 0 used as the reference sample. *p value <0.05, **p value <0.01, ****p value <0.0001, and ns not significant. Boxplots illustrate the distribution of data as described in Fig. 1.

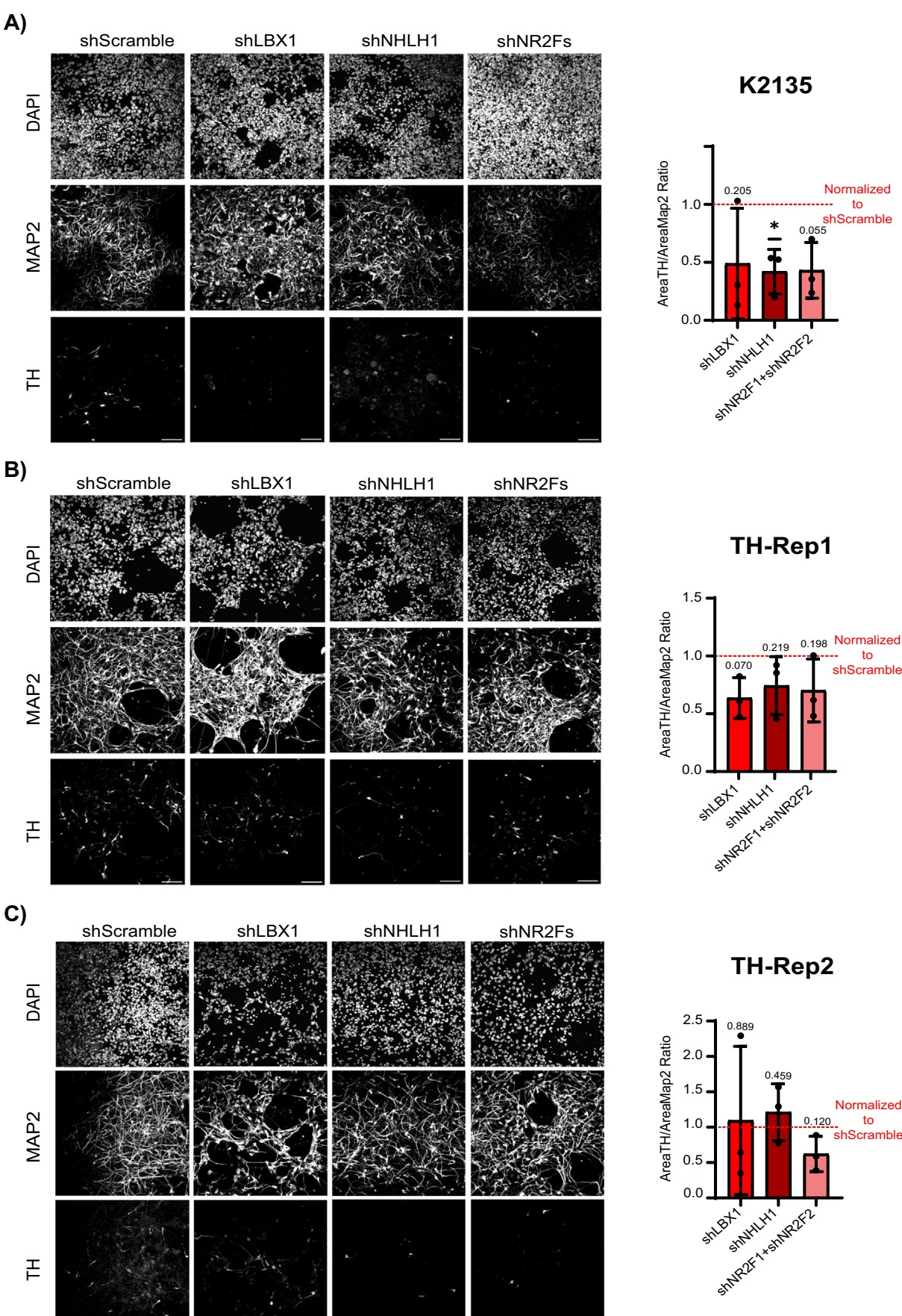

◀　　**Figure EV4.　NHLH1, LBX1, NR2F1/2 are necessary for mDAN differentiation in multiple cell lines.**

(**A–C**) High-content imaging analysis of (**A**) K2135 cell line, (**B**) TH-Rep1 cell line, and (**C**) TH-Rep2 cell line at day 15 of differentiation following late transduction with shRNA lentiviral particles. Cells were stained for the nuclear marker DAPI, the neuronal marker MAP2 and the marker for mDANs TH. Scale bar = 100 μm. Quantification of the TH-stained area over the MAP2 area. Ratios were normalized to the shSCRAMBLE per replicate. Data are representative of 3 independent experiments. Error bars correspond to ±1 standard deviation (SD) from the mean. One sample *t* test was used for statistical analysis, taking 1 as the theoretical mean for TH quantification. *$p$ value <0.05.

