## [Peer Review File · EMBO Reports]

Multimomics analysis identifies novel facilitators of human dopaminergic neuron differentiation

Borja Gomez Ramos, Jochen Ohnmacht, Nikola de Lange, Elena Valceschini, Aurélien Ginolhac, Marie Catillon, Daniele Ferrante, Aleksandar Rakovic, Rashi Halder, Francois Massart, Giuseppe Arena, Paul Antony, Silvia Bolognin, Christine Klein, Roland Krause, Marcel Schulz, Thomas Sauter, Rejko Krueger, and Lasse Sinkkonen

DOI: [10.15252/embr.202357349](https://doi.org/10.15252/embr.202357349)

Corresponding author(s): Lasse Sinkkonen (lasse.sinkkonen@uni.lu)

Review Timeline:

Transfer from Review Commons:	17th Apr 23
Editorial Decision:	10th May 23
Revision Received:	26th Sep 23
Editorial Decision:	13th Nov 23
Revision Received:	17th Nov 23
Accepted:	23rd Nov 23

Editor: Esther Schnapp

Transaction Report: This manuscript was transferred to EMBO reports following peer review at Review Commons.

Review
COMMONS

Review #1

1. Evidence, reproducibility and clarity:

Evidence, reproducibility and clarity (Required)

In this study, Ramos and colleagues defined gene regulatory networks and transcriptional landscape during differentiation of a human iPSC reporter line into dopaminergic neurons. Several omic techniques (RNA-seq, ATAC-seq, chromatin-IP) and modelling (EPIC-DREAM) allowed them to identify putative effectors of dopaminergic differentiation LBX1, NHLH1 and NR2F1/2. Using overexpression and shRNA-mediated knock down experiments, the authors attempted to validate the hits.

This manuscript is very difficult to read and is confusing. The data are interesting, but they need to be presented in a more concise and readable way, in addition to be validated using additional iPSC lines. Below are few comments.

Only relative numbers and mRNA level normalized to control are presented in main figures. This is very confusing because there is no real quantification. Images of cultures to show increased/decreased number of dopaminergic neurons in non-FACS purified cultures following overexpression/knock down should be presented in main figures. It is recommended to add absolute quantification (percent of DAPI) and statistical analysis based on N=3 independent experiments.

There is not data showing that the targets are specifically required for dopaminergic differentiation. One may argue that same targets may be identified and required for differentiation of other neuronal cell types. Hence, hits need to be validated for other neuronal cell types using knock in and shRNA mediated KO.

Based on images shown in figure S4, the effect of rapamycin is very low (no quantification is presented).

Are the neurons generated following overexpression/shRNA-mediated knock down of the three hits functional? Electrophysiological recordings could help.

What other functions are affected in dopaminergic neurons when targets are knocked-down? Is lysosomal activity changed? Is the level of synaptic proteins altered compared to control?

Are the three hits altered in dopaminergic neurons in Parkinson's disease and other

synucleinopathies that could explain dysfunction of dopamine neurons in disease? Nurr1, EN1 and many other genes required for differentiation of dopaminergic neurons from pluripotent stem cells have their expression decreased in Parkinson's. It is expected that the expression of LBX1, NHLH1 and NR2F1/2 would change under disease condition.

2. Significance:

Significance (Required)

Interesting study that needs to be replicated using additional cell lines.

3. How much time do you estimate the authors will need to complete the suggested revisions:

Estimated time to Complete Revisions (Required)

(Decision Recommendation)

Between 3 and 6 months

No

Review #2

1. Evidence, reproducibility and clarity:

Evidence, reproducibility and clarity (Required)

In this manuscript Ramos et al. present a novel and comprehensive transcriptomic and epigenomic profile that identifies a series of key regulators of mDANs differentiation, providing functional validation and characterization of two newly associated TFs: LBX1 or NHLH1. In order to discover key regulators of mDAN differentiation the authors use their previous EPIC-DREM pipeline together with ATAC-seq data for the first time. Then, they focus their attention on those TFs with a more probable regulatory role by performing low input ChIP-seq for H3K27ac leading to the identification of 6 TFs as novel candidate regulators of mDAN differentiation under the control of super-enhancers at day 30 and day 50 of differentiation. In vitro knock down and overexpression of candidate TFs revealed LBX1, NHLH1 as important regulators of DAN differentiation. The authors then interrogate the role of these two TFs through RNA-seq and an Ingenuity Pathway Analysis (IPA)/g:Profiler and proposed regulation of the mature form miR-124 and cholesterol biosynthesis-related genes as the main processes controlled by NHLH1 and LBX1, respectively. Overall, the manuscript is well written, the research laid out in a clear way, and the experiments well thought. The novelty of this study lays in the combination of epigenomic and transcriptomic data at different time points in specific cells during DAN differentiation. I believe the conclusions are supported by the results presented and therefore recommend this paper for publication after addressing some minor points listed below:

****Minor comments:****

1. In page 15 the authors state "the list of 17 TFs was further explored to select the most promising candidates for functional analysis". However, they only named TCF4 and MEIS1 as examples of discarded TFs through literature search. It is not clear which of the remaining 15 TFs were discarded because of a literature search and which were by SE signal cutoff. Clarification is needed.
2. In page 15 the authors state "TFs, HOXB2, LBX1, NHLH1, NR2F1 (also known as COUP-TFI), NR2F2 (also known as COUP-TFII) and SOX4 were found to present the strongest SE signals and most dynamic gene expression profiles" however I could not find the data that corroborate this statement within tables or figures. Authors should provide hard data to support this statement.
3. In supplementary 3, in the IPA analysis some data appear with the warning "#;NUM!" at the z-score. Some explanation should be given and if pertinent, added to the table legend.
4. In methodology, some reagents and techniques appear with a code reference to catalog number and others don't. Please keep it uniform throughout the text.

5. Supplementary table 1 has some TFs highlighted in yellow but there is no legend that explain what the yellow highlight symbolizes. Clarification is needed

****Format suggestions:****

1. For an easier to follow flow between figure 3A and the main text, it would be helpful if NR2F1 and NR2F2 graphs in Figure 3A appeared next to each other or one above the other.

2. Supplementary table 2.

- a. Data is presented in a confused way. For example, the Top20_TFs_EPIC-DREM is presented as a list of names without divisions of type node or scoring annotations. It would be more informative and easy to follow if proper labeling and scoring is given within this spreadsheet without the necessity of navigating sup.table1 in parallel.

- b. it would be preferable to have an extra sheet showing the comparison between both data sets (SE and EPICDREAM) before providing a final list of relevant TFs.

2. Significance:

Significance (Required)

General assessment:

Overall, the manuscript is well written, the research laid out in a clear way, and the experiments well thought. The novelty of this study lays in the combination of epigenomic and transcriptomic data at different time points in specific cells during DAN differentiation and description of new roles in DAN differentiation for two TF: LBX1 or NHLH1.

Limitations:

One limitation, than the authors themselves mentioned, is the possibility that promising candidate TFs involved in mDAN differentiation are discarded or not taken in account by the EPIC-DREAM algorithm.

Audience:

This manuscript focuses on factors involved in mDAN differentiation which targets a highly specific audience however their multiomic and functional methodology might attract broader audiences looking to apply similar pipelines and/or experimentation in different areas of research.

My field of expertise:

I am a geneticist and neuroscientist with expertise in molecular biology and epigenomics focused on age related neurodegenerative disorders.

Recommendation:

I believe the conclusions are supported by the results presented and therefore recommend this paper for publication after addressing some minor points.

3. How much time do you estimate the authors will need to complete the suggested revisions:

Estimated time to Complete Revisions (Required)

(Decision Recommendation)

Less than 1 month

No

Revision Plan

Manuscript number: RC-2023-01862

Corresponding author(s): Lasse, Sinkkonen

1. General Statements [optional]

This section is optional. Insert here any general statements you wish to make about the goal of the study or about the reviews.

In our manuscript we have aimed to take an unbiased and data-driven high-throughput approach for identification of transcription factors important for dopaminergic neuron differentiation via repeated, combined transcriptomics and epigenomics measurements. We also provide the research community with an extensive dataset enabling further studies on dopaminergic neurons beyond the scope of a single manuscript. We validate identified transcription factors not previously recognized being involved in mDAN differentiation. While we believe our approach is powerful in unbiased identification of central regulators, it does not focus only on factors that are unique for dopaminergic neurons. Importantly, the ranking of transcription factors is based on the epigenomic data of the target genes, rather than expression of transcription factors themselves. We have aimed for the genome-wide identification of pathways controlled by the identified transcription factors, for example through transcriptome analysis.

For practical reasons, to gain the sufficient depth of data to accomplish our aim, only one iPSC line was used for the initial data generation. However, we fully agree on the need for validation of the key findings and overall gene expression profiles in additional independent cell lines. Please find below our detailed point-by-point plan on addressing the reviewers' comments.

2. Description of the planned revisions

Insert here a point-by-point reply that explains what revisions, additional experimentations and analyses are planned to address the points raised by the referees.

Reviewer #1 (Evidence, reproducibility and clarity (Required)):

In this study, Ramos and colleagues defined gene regulatory networks and transcriptional landscape during differentiation of a human iPSC reporter line into dopaminergic neurons. Several omic techniques (RNA-seq, ATAC-seq, chromatin-IP) and modelling (EPIC-DREAM) allowed them to identify putative effectors of dopaminergic differentiation LBX1, NHLH1 and NR2F1/2. Using overexpression and shRNA-mediated knock down experiments, the authors attempted to validate the hits.

This manuscript is very difficult to read and is confusing. The data are interesting, but they need to be presented in a more concise and readable way, in addition to be validated using additional iPSC lines. Below are few comments.

Revision Plan

We thank Reviewer1 for taking the time to evaluate our manuscript and for providing valuable feedback towards improving it further. We were happy to read that the Reviewer1 found the study interesting with only a few caveats. Here we will outline a detailed plan to address those limitations.

In the revised manuscript we will do our best to improve the readability of the manuscript. However, since Reviewer2 has found that the manuscript is "*well written, the research laid out in a clear way, and the experiments well thought*", it is somewhat difficult for us to identify the exact changes to introduce. Perhaps these are related to field-specific vocabulary or methodology, which we will aim to make more readable for broader audience.

We agree on the concern of Reviewer1 that different human iPSC lines can show significant variability due to their individual genetic backgrounds. We have observed differences in the rate of neuronal differentiation, depending on the iPSC line, and transcriptomic analysis reveals hundreds of differentially expressed genes between independent iPSC lines. Still, in a case of a single healthy donor, we don't expect an intra-individual variability to alter conclusions regarding key regulators of fundamental processes such as differentiation. To carry out our multi-omic analysis in the sufficient depth that we have applied and using only purified dopaminergic neurons with a TH-mCherry-reporter inserted using genome editing, it was (also budget-wise) not considered to include multiple independent iPSC lines for the entire panel of experiments (as our ambition was not to characterize a specific mutation). However, to address this point, we have generated a second iPSC line from a healthy donor with TH-mCherry-reporter inserted through genome editing. To address the concerns regarding variability between different human iPSC lines we plan to:

- 1) Perform transcriptomic profiling of the second TH-mCherry-reporter line at selected time points of dopaminergic neuron differentiation to confirm the similarity of changes in cell identity at transcriptome level.
- 2) Perform TH staining upon LBX1 or NHLH1 knock-down in additional iPSC lines following dopaminergic neuron differentiation, to confirm their effect on differentiation across iPSC lines. To do this we will apply the Yokogawa high content image analysis that has been recently established in our laboratories. This will also be related to the next point regarding microscopy images of the dopaminergic neuron differentiation and the effect of transcription factors on this.

Only relative numbers and mRNA level normalized to control are presented in main figures. This is very confusing because there is no real quantification. Images of cultures to show increased/decreased number of dopaminergic neurons in non-FACS purified cultures following overexpression/knock down should be presented in main figures. It is recommended to add absolute quantification (percent of DAPI) and statistical analysis based on N=3 independent experiments.

Thank you for raising this point. We are happy to clarify the quantification of dopaminergic neuron numbers and mRNA levels. All quantifications of dopaminergic neuron numbers were based on

Revision Plan

the mCherry reporter inserted in the TH locus through genome editing and expressed together with endogenous TH. While mCherry can be detected using microscopy (as shown in Figure 1), the signal is significantly weaker than what can be achieved through antibody staining and quantitative analysis is therefore much more accurate when systematically performed using FACS analysis and controlled by using a cell line without the mCherry reporter. Moreover, the approach is direct and not dependent on antibody specificity. Therefore, all quantifications of dopaminergic neuron numbers in the manuscript were performed using FACS.

Most *in vitro* cell differentiation protocols show variability in their efficiency between independent experiments, which is typically reflected as variable expression levels of the different marker genes (Grancharova et al. 2021). This is also true for dopaminergic neuron differentiation and in our experiments the number of obtained dopaminergic neurons can vary between 5-20% while differentiations performed in parallel as part of the same experiment are typically very similar. Summarizing absolute numbers between independent experiments can lead to large variation while the relative effect of perturbation is reproducible. Therefore, our results are presented as relative changes in dopaminergic neuron numbers and mRNA levels.

Nevertheless, to increase confidence in the impact of NHLH1 and LBX1 on dopaminergic neuron differentiation, we propose, as already described above, to perform TH staining upon LBX1 or NHLH1 knock-down in additional iPSC lines following dopaminergic neuron differentiation. To visualize the observed impact on differentiation.

Based on images shown in figure S4, the effect of rapamycin is very low (no quantification is presented).

We apologize for the unclear Figure Legend for Figure S4 that did not specify what is visualized in the image. The images represent the transduction efficiency of the neurons based on the GFP reporter co-expressed with the short hairpin constructs. The mCherry levels, that are quantified in Figure 6G, are not visualized in these images. We will correct the Figure Legend accordingly. As mentioned in the last sentence on page 20 of the manuscript (referring to Supplementary Figure S4), rapamycin did not induce similar level of reduction in cell numbers as LBX1 knock-down alone did.

Are the three hits altered in dopaminergic neurons in Parkinson's disease and other synucleinopathies that could explain dysfunction of dopamine neurons in disease? Nurr1, EN1 and many other genes required for differentiation of dopaminergic neurons from pluripotent stem cells have their expression decreased in Parkinson's. It is expected that the expression of LBX1, NHLH1 and NR2F1/2 would change under disease condition.

We have investigated the expression of LBX1, NHLH1 and NR2F1/2 using recent meta-analysis of post-mortem brain tissue transcriptomes of Parkinson's disease patients (Tranchevent, Halder, & Glaab, 2023). None of these TFs was found to be dysregulated in Parkinson's disease patients. This is consistent with the fact that the expression of these factors is not restricted only to A9

Revision Plan

midbrain dopaminergic neurons that are primarily degenerating in Parkinson's disease but can be detected also in several other types of neurons (please see also our response in section 4).

However, NHLH1 expression is reduced in dopaminergic neurons derived from iPSCs of Parkinson's disease patients carrying a LRRK2-G2019S mutation based on our published single cell RNA-seq data (Walter et al., 2021).

Beyond this, our results implicate NHLH1 in the regulation of miR-124, which in turn has been found to be downregulated in Parkinson's disease patients and neuroprotective in different animal models of Parkinson's disease (Angelopoulou, Paudel, & Piperi, 2019; Saraiva, Paiva, Santos, Ferreira, & Bernardino, 2016; Yang, Li, Yang, Guo, & Li, 2021; Zhang et al., 2022). Similarly, a recent analysis of single nuclei RNA-seq of midbrains from Parkinson's disease patients, showed that targets of NR2F2 were enriched in the vulnerable dopaminergic neuron population, promoting neurodegeneration (Kamath et al., 2022). Indicating the involvement of the pathway in disease progression without a change in transcription factor expression.

Finally, a polymorphism in NHLH1 locus (rs2147472) is associated with schizophrenia while a polymorphism in LBX1 locus (rs12242050) is associated with Parkinson's disease, suggesting further involvement of these genes in disease risk.

We propose to include these findings in the revised manuscript and discuss them in the context of the current literature.

Reviewer #1 (Significance (Required)):

Interesting study that needs to be replicated using additional cell lines.

We would like to thank the reviewer for this positive conclusion and plan to address the key concerns using additional iPSC lines for transcriptome profiling and knock-down experiments.

Reviewer #2 (Evidence, reproducibility and clarity (Required)):

In this manuscript Ramos et al. present a novel and comprehensive transcriptomic and epigenomic profile that identifies a series of key regulators of mDANs differentiation, providing functional validation and characterization of two newly associated TFs: LBX1 or NHLH1. In order to discover key regulators of mDAN differentiation the authors use their previous EPIC-DREM pipeline together with ATAC-seq data for the first time. Then, they focus their attention on those TFs with a more probable regulatory role by performing low input ChIP-seq for H3K27ac leading to the identification of 6 TFs as novel candidate regulators of mDAN differentiation under the control of super-enhancers at day 30 and day 50 of differentiation. In vitro knock down and overexpression of candidate TFs revealed LBX1, NHLH1 as important regulators of DAN differentiation. The authors then interrogate the role of these two TFs through RNA-seq and an Ingenuity Pathway Analysis (IPA)/g:Profiler and proposed regulation of the mature form miR-124

Revision Plan

and cholesterol biosynthesis-related genes as the main processes controlled by NHLH1 and LBX1, respectively.

Overall, the manuscript is well written, the research laid out in a clear way, and the experiments well thought. The novelty of this study lays in the combination of epigenomic and transcriptomic data at different time points in specific cells during DAN differentiation. I believe the conclusions are supported by the results presented and therefore recommend this paper for publication after addressing some minor points listed below:

We would like to thank the reviewer for the detailed and overall positive evaluation of our work. We are grateful for the suggestions for improvements and below we detail our plan for addressing them.

Minor comments:

1. In page 15 the authors state "the list of 17 TFs was further explored to select the most promising candidates for functional analysis". However, they only named TCF4 and MEIS1 as examples of discarded TFs through literature search. It is not clear which of the remaining 15 TFs were discarded because of a literature search and which were by SE signal cutoff. Clarification is needed.

We will add clarification statements here, providing more evidence for the selection of our candidates. For that, we will add a supplementary figure showing the locus and expression of the 11 TFs that were not selected.

2. In page 15 the authors state "TFs, HOXB2, LBX1, NHLH1, NR2F1 (also known as COUP-TFI), NR2F2 (also known as COUP-TFII) and SOX4 were found to present the strongest SE signals and most dynamic gene expression profiles" however I could not find the data that corroborate this statement within tables or figures. Authors should provide hard data to support this statement.

With the clarification from point 1, point 2 will also be answered for a clear description and criteria of our selection.

3. In supplementary 3, in the IPA analysis some data appear with the warning "#iNUM!" at the z-score. Some explanation should be given and if pertinent, added to the table legend.

Sorry for not clarifying that in the dataset. That term is produced when IPA cannot predict the Z-score and it is represented in our bar graphs in grey (see Figure 6B). We will add that information to the table header.

4. In methodology, some reagents and techniques appear with a code reference to catalog number and others don't. Please keep it uniform throughout the text.

Revision Plan

In this study, we performed most of the techniques using kits which contained all necessary reagents for it. We will better clarify which reagents were provided by the manufacturer and which ones were additional to the kits.

5. Supplementary table 1 has some TFs highlighted in yellow but there is no legend that explain what the yellow highlight symbolizes. Clarification is needed

This is an error and there should not be any TF highlighted in yellow. We apologize for the inconvenience. The highlights will be removed from the revised tables.

Format suggestions:

1. For an easier to follow flow between figure 3A and the main text, it would be helpful if NR2F1 and NR2F2 graphs in Figure 3A appeared next to each other or one above the other.

We will follow the recommendation from the reviewer, and we will change the order of the TFs in the figure to have both NR2F TFs next to each other.

2. Supplementary table 2.

a. Data is presented in a confused way. For example, the Top20_TFs_EPIC-DREM is presented as a list of names without divisions of type node or scoring annotations. It would be more informative and easy to follow if proper labeling and scoring is given within this spreadsheet without the necessity of navigating sup.table1 in parallel.

b. it would be preferable to have an extra sheet showing the comparison between both data sets (SE and EPICDREAM) before providing a final list of relevant TFs.

To the existing table containing the TFs controlled by SE, we are going to add the information regarding EPIC-DREM, namely, the rank those TFs got in each node together with their median ranking, and their best rank across nodes. That will give a good overview on how they look in both analyses. With this approach, some of the TFs controlled by SE will have no information regarding EPIC-DREM because their motif is not known according to the Jaspar database.

Reviewer #2 (Significance (Required)):

-General assessment:

Overall, the manuscript is well written, the research laid out in a clear way, and the experiments well thought. The novelty of this study lays in the combination of epigenomic and transcriptomic data at different time points in specific cells during DAn differentiation and description of new roles in DAn differentiation for two TF: LBX1 or NHLH1.

We thank the Reviewer2 for this assessment.

-Limitations:

Revision Plan

One limitation, than the authors themselves mentioned, is the possibility that promising candidate TFs involved in mDAN differentiation are discarded or not taken in account by the EPIC-DREAM algorithm.

We agree with this limitation and will make all the data available for the larger research community to use for follow-up work. Importantly, the data can be easily used for re-analysis using the same pipeline when improved databases become available.

-Audience:

This manuscript focuses on factors involved in mDAN differentiation which targets a highly specific audience however their multiomic and functional methodology might attract broader audiences looking to apply similar pipelines and/or experimentation in different areas of research.

-My field of expertise:

I am a geneticist and neuroscientist with expertise in molecular biology and epigenomics focused on age related neurodegenerative disorders.

-Recommendation:

I believe the conclusions are supported by the results presented and therefore recommend this paper for publication after addressing some minor points.

3. Description of the revisions that have already been incorporated in the transferred manuscript

Please insert a point-by-point reply describing the revisions that were already carried out and included in the transferred manuscript. If no revisions have been carried out yet, please leave this section empty.

No changes were introduced so far.

4. Description of analyses that authors prefer not to carry out

Please include a point-by-point response explaining why some of the requested data or additional analyses might not be necessary or cannot be provided within the scope of a revision. This can be due to time or resource limitations or in case of disagreement about the necessity of such additional data given the scope of the study. Please leave empty if not applicable.

Below we will provide a point-by-point response to concerns raised by the reviewers that we believe are outside of the scope of our study.

There is not data showing that the targets are specifically required for dopaminergic differentiation. One may argue that same targets may be identified and required for differentiation

Revision Plan

of other neuronal cell types. Hence, hits need to be validated for other neuronal cell types using knock in and shRNA mediated KO.

The novelty of this study resides in the use of epigenomic signatures to predict TF activity across differentiation and couple those predictions with the transcriptional changes occurring during this process to identify the TFs responsible for most of the transcriptional changes observed. Therefore, although our focus were TFs important for establishing cell identity, we did not select TFs with a selective/exclusive expression in these cells, namely, cell identity TFs. This gives another perspective regarding TF activity and their relevance for cellular processes like differentiation.

We believe the TFs identified in our study are likely to be involved in regulation in several other neuronal subtypes. There is a wide range of neuronal subtypes and selection and establishment of some of the those for testing of our factors seems biased but also outside of the scope of this study.

Are the neurons generated following overexpression/shRNA-mediated knock down of the three hits functional? Electrophysiological recordings could help.

What other functions are affected in dopaminergic neurons when targets are knocked-down? Is lysosomal activity changed? Is the level of synaptic proteins altered compared to control?

In order not to bias our approach towards particular phenotypes by selected analysis such as electrophysiological measurements or lysosomal activity assays, we performed a transcriptional profiling upon TF depletion for our selected candidates. Our transcriptional profiling highlighted the main pathways affected by the TFs and they are presented and discussed in our study. We exploit these data to find the processes controlled by our TFs that help to define dopaminergic neuron cell identity. We discussed them and tested the role of mTOR signaling and miR-124 as targets of our TFs. The results from the RNA-seq analysis did not indicate direct regulation of synaptic or lysosomal activity, and therefore we find such analysis to be outside of the scope of our study.

Moreover, since the knock-down of our candidate TFs is in general inhibiting dopaminergic differentiation, studying the dopaminergic neurons remaining after a knock-down risks focusing on cells that have either partially or completely escaped the knock-down. Thereby influencing the value of detailed analysis of their functionality.

References:

Grancharova, T., Gerbin, K.A., Rosenberg, A.B. et al. A comprehensive analysis of gene expression changes in a high replicate and open-source dataset of differentiating hiPSC-derived cardiomyocytes. *Sci Rep* 11, 15845 (2021). <https://doi.org/10.1038/s41598-021-94732-1>

Angelopoulou, E., Paudel, Y. N., & Piperi, C. (2019). miR-124 and Parkinson's disease: A biomarker with therapeutic potential. *Pharmacological Research*, 150. <https://doi.org/10.1016/J.PHRS.2019.104515>

Revision Plan

Kamath, T., Abdulraouf, A., Burris, S. J., Langlieb, J., Gazestani, V., Nadaf, N. M., ... Macosko, E. Z. (2022). Single-cell genomic profiling of human dopamine neurons identifies a population that selectively degenerates in Parkinson's disease. *Nature Neuroscience*, 25(5), 588–595. <https://doi.org/10.1038/S41593-022-01061-1>

Saraiva, C., Paiva, J., Santos, T., Ferreira, L., & Bernardino, L. (2016). MicroRNA-124 loaded nanoparticles enhance brain repair in Parkinson's disease. *Journal of Controlled Release : Official Journal of the Controlled Release Society*, 235, 291–305. <https://doi.org/10.1016/J.JCONREL.2016.06.005>

Tranchevent, L. C., Halder, R., & Glaab, E. (2023). Systems level analysis of sex-dependent gene expression changes in Parkinson's disease. *Npj Parkinson's Disease* 2023 9:1, 9(1), 1–16. <https://doi.org/10.1038/s41531-023-00446-8>

Walter, J., Bolognin, S., Poovathingal, S. K., Magni, S., Gérard, D., Antony, P. M. A., ... Schwamborn, J. C. (2021). The Parkinson's-disease-associated mutation LRRK2-G2019S alters dopaminergic differentiation dynamics via NR2F1. *Cell Reports*, 37(3). <https://doi.org/10.1016/J.CELREP.2021.109864>

Yang, Y., Li, Y., Yang, H., Guo, J., & Li, N. (2021). Circulating MicroRNAs and Long Non-coding RNAs as Potential Diagnostic Biomarkers for Parkinson's Disease. *Frontiers in Molecular Neuroscience*, 14, 28. <https://doi.org/10.3389/FNMOL.2021.631553/BIBTEX>

Zhang, F., Yao, Y., Miao, N., Wang, N., Xu, X., & Yang, C. (2022). Neuroprotective effects of microRNA 124 in Parkinson's disease mice. *Archives of Gerontology and Geriatrics*, 99. <https://doi.org/10.1016/J.ARCHGER.2021.104588>

Dear Dr. Sinkkonen,

Thank you for the submission of your manuscript and proposed revision plan to EMBO reports. I am very sorry for my delayed reply. First I was waiting for input from an advisor I contacted, and then I was traveling all of last week.

After discussing your study and revision plan with my colleagues here, we decided that we would like to invite you to address all referee comments as you suggest. However, we would also ask for some more functional data after overexpression and knockdown of LBX1, NHLH1 and NR2F1/2, as referee 1 suggests. Please let me know in case you would like to discuss this further.

I would thus like to invite you to revise your manuscript with the understanding that the referee concerns must be fully addressed and their suggestions taken on board. Please address all referee concerns in a complete point-by-point response. Acceptance of the manuscript will depend on a positive outcome of a second round of review. It is EMBO reports policy to allow a single round of major revision only and acceptance or rejection of the manuscript will therefore depend on the completeness of your responses included in the next, final version of the manuscript.

We realize that it is difficult to revise to a specific deadline. In the interest of protecting the conceptual advance provided by the work, we recommend a revision within 3 months (10th Aug 2023). Please discuss the revision progress ahead of this time with the editor if you require more time to complete the revisions.

- 1) A data availability section providing access to data deposited in public databases is missing. If you have not deposited any data, please add a sentence to the data availability section that explains that.
- 2) Your manuscript contains statistics and error bars based on $n=2$. Please use scatter blots in these cases. No statistics should be calculated if $n=2$.

3) We replaced Supplementary Information with Expanded View (EV) Figures and Tables that are collapsible/expandable online. A maximum of 5 EV Figures can be typeset. EV Figures should be cited as 'Figure EV1, Figure EV2' etc... in the text and their respective legends should be included in the main text after the legends of regular figures.

5) a complete author checklist, which you can download from our author guidelines . Please insert information in the checklist that is also reflected in the manuscript. The completed author checklist will also be part of the RPF.

6) Please note that all corresponding authors are required to supply an ORCID ID for their name upon submission of a revised manuscript (). Please find instructions on how to link your ORCID ID to your account in our manuscript tracking system in our Author guidelines

7) Before submitting your revision, primary datasets produced in this study need to be deposited in an appropriate public database (see <https://www.embopress.org/page/journal/14693178/authorguide#datadeposition>). Please remember to provide a

reviewer password if the datasets are not yet public. The accession numbers and database should be listed in a formal "Data Availability" section placed after Materials & Method (see also <https://www.embopress.org/page/journal/14693178/authorguide#datadeposition>). Please note that the Data Availability Section is restricted to new primary data that are part of this study. * Note - All links should resolve to a page where the data can be accessed. *

- the name of the statistical test used to generate error bars and P values,
- the number (n) of independent experiments (please specify technical or biological replicates) underlying each data point,
- the nature of the bars and error bars (s.d., s.e.m.),
- If the data are obtained from n Program fragment delivered error ``Can't locate object method "less" via package "than" (perhaps you forgot to load "than"?) at //ejpvfs23/sites23b/embor_www/letters/embor_decision_rc_revise_and_rereview.txt line 56.' 2, use scatter blots showing the individual data points.

I look forward to seeing a revised form of your manuscript when it is ready.

Yours sincerely,

Authors' Response to Reviewers

25th of September 2023

Manuscript number: EMBOR-2023-57349V1 [RC-2023-01862]**Corresponding author(s):** Sinkkonen, Lasse

1. General Statements

We would like to thank you for considering our manuscript for publication in EMBO Reports and appreciate the very useful feedback from the reviewers. We have executed the previously proposed revision plan, used data from additional cell lines to strengthen our conclusions, and included a new main figure investigating the role of NR2F1/2 in regulation of neuronal activity. To reflect this change and to avoid diluting the manuscript title, we propose a new title: "*Multi-omics analysis identifies novel central regulators of human dopaminergic neuron differentiation*". Please find below our detailed point-by-point response to reviewers. All changes in the manuscript text are in red font. We believe the manuscript has significantly improved and provides useful insights for the readership of EMBO Reports.

2. Point-by-point response to reviewers

Reviewer #1 (Evidence, reproducibility and clarity (Required)):

In this study, Ramos and colleagues defined gene regulatory networks and transcriptional landscape during differentiation of a human iPSC reporter line into dopaminergic neurons. Several omic techniques (RNA-seq, ATAC-seq, chromatin-IP) and modelling (EPIC-DREAM) allowed them to identify putative effectors of dopaminergic differentiation LBX1, NHLH1 and NR2F1/2. Using overexpression and shRNA-mediated knock down experiments, the authors attempted to validate the hits.

This manuscript is very difficult to read and is confusing. The data are interesting, but they need to be presented in a more concise and readable way, in addition to be validated using additional iPSC lines. Below are few comments.

Thank you to Reviewer 1 for the very useful suggestions that have helped us to significantly improve the manuscript. We have reorganized panels in many of the figures to improve the flow between text and figures, generated new schematic representations in Figures 1A, 1E, 3A, and 4A, and provide a graphical abstract of the main findings. We have also attempted to improve the clarity of the newly inserted sentences while keeping in mind that Reviewer 2 found text "*well written*", and therefore avoiding too many changes. We have added new Expanded View Figures 1B, 2, 3, and 4 to support and ease the readability of the main figures.

We agree with the reviewer that independent iPSC lines can show significant variability and agree on the importance of validating findings in different cell lines through independent approaches and experiments. To this end, we have:

- 1) Generated a second independent TH-reporter cell line (TH-Rep2) and used it for time-series RNA-seq analysis to confirm the induction of various general and mDAN-specific

marker genes as shown in the new Expanded View Figure 1B (please see the figure below).

- 2) Analyzed the expression profiles of the candidate transcription factors from the above RNA-seq data from the second independent TH-reporter cell line as shown in the new Expanded View Figure 3A (please see the figure below).
- 3) Obtained a permission from the Parkinson's Progression Markers Initiative (PPMI) to leverage their RNA-seq data from differentiation of 95 independent iPSC lines towards mDANs, to further strengthen the conclusion regarding the induced expression of the candidate transcription factors in mDANs. The data from all 95 independent cell lines are plotted in the new Expanded View Figure 3B, confirming increased expression of all candidate transcription factors in particular during dopaminergic neurogenesis (please see the figure below).
- 4) Performed high content imaging for TH-positive neurons upon knock-down of the selected candidate transcription factors in four independent iPSC lines, two non-reporter lines (17608 and K2135) and two TH-reporter lines (TH-Rep1 and TH-Rep2). The results are shown in the new main Figure 4D, the new Expanded View Figure 4, and the new Supplementary Figure 2 that are shown and discussed in more detail under the next point.

The above mentioned new Expanded View Figure 1B and Expanded View Figure 3 can be found also here:

Expanded View Figure 1

A)

B)

Expanded View Figure 3

Only relative numbers and mRNA level normalized to control are presented in main figures. This is very confusing because there is no real quantification. Images of cultures to show increased/decreased number of dopaminergic neurons in non-FACS purified cultures following overexpression/knock down should be presented in main figures. It is recommended to add absolute quantification (percent of DAPI) and statistical analysis based on N=3 independent experiments.

We agree with the reviewer on the importance of applying independent methods to strengthen conclusions. To test the impact of LBX1, NHLH1, and NR2F1/2 knock-downs on mDAN differentiation independently of FACS analysis of mCherry reporter, we have performed high content imaging to acquire >1000 images across ≥ 3 independent differentiations from 4 independent cell lines to do quantitative analysis of TH (for mDANs), MAP2 (for neurons), and DAPI (for nuclei) area. The relative quantifications and representative stainings are shown in main Figure 4D for cell line 17608 (please see below) and Expanded View Figure 4 for cell lines

K2135, TH-Rep1, and TH-Rep2 (please see below). Observing the ratio of TH area within the MAP2 area confirmed a trend of reduced number of TH-positive neurons in three out of the four tested cell lines, with the quantifications from 17608 cell line showing highest significance of reduction for all tested transcription factors. While for TH-Rep2 cells the results were too variable to draw clear conclusions.

The observed variation between independent differentiation experiments is not unexpected. As described in the original revision plan, most *in vitro* cell differentiation protocols show variability in their efficiency between independent experiments (Grancharova et al. 2021) and this has been recently shown also specifically for mDAN differentiation (Bressan et al. 2023). Summarizing absolute numbers between independent experiments can lead to large variation while the relative effect of perturbation is reproducible. This is also the reason why we have continued to present our data as relative numbers. However, in addition to quantifications, in the revised manuscript we now also present microscopy images of the transduced and stained neurons, and as before, our quantifications are based on $N \geq 3$ independent differentiations.

Figure 4

Expanded View Figure 4

In addition to the knock-down experiments we have aimed to investigate the impact of LBX1 and NHLH1 overexpression on mDAN differentiation using high content imaging. Independent overexpression experiments were performed, followed by analysis of TH, MAP2, and DAPI stainings as described above. Similarly to the FACS analysis, a trend of increased TH area compared to nuclei area upon NHLH1 overexpression was observed (please see the figure below). However, investigation of the transduction efficiency based on GFP signal revealed that the control cells receiving shScramble did not exhibit any GFP expression (please see the figure below). Therefore, we could not consider the results properly controlled and have not included the results in the revised manuscript. However, we wanted to bring the data to the attention of the reviewer.

Are the neurons generated following overexpression/shRNA-mediated knock down of the three hits functional? Electrophysiological recordings could help.

What other functions are affected in dopaminergic neurons when targets are knocked-down? Is lysosomal activity changed? Is the level of synaptic proteins altered compared to control?

Finally, in addition to the detailed analysis of targets of LBX1 and NHLH1 presented in Figures 5 and 6, and already included in the earlier version of the manuscript, we have now performed RNA-seq analysis upon knock-down of NR2F1 and NR2F2 to identify their putative target genes as shown in the new main Figure 7A (please see below). Pathway enrichment analysis of the RNA-seq data pointed to an important role of NR2F1/2 in the control of synaptic activity and related processes as shown in the new main Figure 7B (please see below). To investigate this putative function of NR2F1/2 in neuronal activity, we have performed multielectrode array analysis of electrophysiological activity during mDAN differentiation upon the knock-down of NR2F1/2 as well as LBX1 and NHLH1 (new main Figure 7C-G, please see below). Consistent with their involvement in mDAN differentiation, each knock-down led to decreased mean firing rate, but the effect was clearly strongest for NR2F1/2 where many electrodes did not report any firing in the presence of the shRNA expression. Indicating an important involvement of these factors in regulation of synaptic activity.

Figure 7

Based on images shown in figure S4, the effect of rapamycin is very low (no quantification is presented).

We apologize for the unclear Figure Legend for Figure S4 that did not specify what is visualized in the image. The images represent the transduction efficiency of the neurons based on the GFP reporter co-expressed with the short hairpin constructs. The mCherry levels, that are quantified in Figure 6G, are not visualized in these images. We have corrected the Figure Legend accordingly. As mentioned in the last sentence on page 12 of the manuscript (referring to Supplementary Figure S4), rapamycin did not induce similar level of reduction in cell numbers as LBX1 knock-down alone did.

Are the three hits altered in dopaminergic neurons in Parkinson's disease and other synucleinopathies that could explain dysfunction of dopamine neurons in disease? Nurr1, EN1 and many other genes required for differentiation of dopaminergic neurons from pluripotent stem cells have their expression decreased in Parkinson's. It is expected that the expression of LBX1, NHLH1 and NR2F1/2 would change under disease condition.

We have investigated the expression of LBX1, NHLH1 and NR2F1/2 using recent meta-analysis of post-mortem brain tissue transcriptomes of Parkinson's disease patients (Tranchevent, Halder, & Glaab, 2023). None of these TFs was found to be dysregulated in Parkinson's disease patients. This is consistent with the fact that the expression of these factors is not restricted only to A9 midbrain dopaminergic neurons that are primarily degenerating in Parkinson's disease but can be detected also in several other types of neurons.

However, our results implicate NHLH1 in the regulation of miR-124, which in turn has been found to be downregulated in Parkinson's disease patients and neuroprotective in different animal models of Parkinson's disease (Angelopoulou, Paudel, & Piperi, 2019; Saraiva, Paiva, Santos, Ferreira, & Bernardino, 2016; Yang, Li, Yang, Guo, & Li, 2021; Zhang et al., 2022).

Similarly, a recent analysis of single nuclei RNA-seq of midbrains from Parkinson's disease patients, showed that targets of NR2F2 were enriched in the vulnerable dopaminergic neuron population, promoting neurodegeneration (Kamath et al., 2022). Indicating the involvement of the pathway in disease progression without a change in transcription factor expression.

Finally, a polymorphism in NHLH1 locus (rs2147472) is associated with schizophrenia while a polymorphism in LBX1 locus (rs12242050) is associated with Parkinson's disease, suggesting further involvement of these genes in disease risk.

We have included these references in different parts of the Discussion, in particular in the new last paragraph of the Discussion on page 15 of the revised manuscript.

Reviewer #1 (Significance (Required)):

Interesting study that needs to be replicated using additional cell lines.

We would like to thank the reviewer for this positive conclusion and believe we have addressed these concerns through the new experiments and analysis included in the revised version of the manuscript.

Reviewer #2 (Evidence, reproducibility and clarity (Required)):

In this manuscript Ramos et al. present a novel and comprehensive transcriptomic and

epigenomic profile that identifies a series of key regulators of mDANs differentiation, providing functional validation and characterization of two newly associated TFs: LBX1 or NHLH1. In order to discover key regulators of mDAN differentiation the authors use their previous EPIC-DREM pipeline together with ATAC-seq data for the first time. Then, they focus their attention on those TFs with a more probable regulatory role by performing low input ChIP-seq for H3K27ac leading to the identification of 6 TFs as novel candidate regulators of mDAN differentiation under the control of super-enhancers at day 30 and day 50 of differentiation. In vitro knock down and overexpression of candidate TFs revealed LBX1, NHLH1 as important regulators of DAN differentiation. The authors then interrogate the role of these two TFs through RNA-seq and an Ingenuity Pathway Analysis (IPA)/g:Profiler and proposed regulation of the mature form miR-124 and cholesterol biosynthesis-related genes as the main processes controlled by NHLH1 and LBX1, respectively.

Overall, the manuscript is well written, the research laid out in a clear way, and the experiments well thought. The novelty of this study lays in the combination of epigenomic and transcriptomic data at different time points in specific cells during DAN differentiation. I believe the conclusions are supported by the results presented and therefore recommend this paper for publication after addressing some minor points listed below:

We would like to thank Reviewer 2 for the detailed and overall positive evaluation of our work. We are grateful for the suggestions that have significantly improved the manuscript.

Minor comments:

1. In page 15 the authors state "the list of 17 TFs was further explored to select the most promising candidates for functional analysis". However, they only named TCF4 and MEIS1 as examples of discarded TFs through literature search. It is not clear which of the remaining 15 TFs were discarded because of a literature search and which were by SE signal cutoff. Clarification is needed.

Please see the response below.

2. In page 15 the authors state "TFs, HOXB2, LBX1, NHLH1, NR2F1 (also known as COUP-TFI), NR2F2 (also known as COUP-TFII) and SOX4 were found to present the strongest SE signals and most dynamic gene expression profiles" however I could not find the data that corroborate this statement within tables or figures. Authors should provide hard data to support this statement.

We agree with the reviewer that the selection of the candidate genes was not well explained and lacked details. To provide more information, we have generated a new Expanded View Figure 2 (please see below) containing the 11 super-enhancer harboring loci, that together with the 6 loci already shown in Figure 3C, make up the 17 loci included in the top 20 ranked TFs from EPIC-DREM and shown in Figure 3B. The new figure contains the H3K27ac ChIP-seq signals and ATAC-seq signals from neuronal precursors and dopaminergic neurons. In addition, the expression profiles of each TF across differentiation are plotted. Comparison of the loci

across the two figures should give a better overview of the observed super-enhancer signals and how prominent the selected candidates are in comparison to others. These data can also help other investigators working on TFs we have not studied here .

In addition, we have provided the full list of the TFs that were excluded because they are already associated with dopaminergic differentiation and functions in the literature. These are listed, along with the relevant references, in a new sentence on page 6 of the revised manuscript:

“TCF4, MEIS1, MEIS2, FOS, JUND, RARA, SP3, and TP53 have already been associated to mDAN subset specification, dopaminergic system formation, mDAN differentiation, and regulation of mDAN specific genes, and consequently were not included in follow-up experiments (Lyu et al, 2020; Mesman et al, 2021; Jiang et al, 2015; Wang & Bannon, 2005; Podleśny-Drabiniok et al, 2017; Agoston et al, 2014; Engele & Schilling, 1996). From the remaining nine TFs, HOXB3 was excluded since the accessibility at the locus in mDANs was centered on the adjacent HOXB2 instead (Figure EV2). Similarly, RFX2 and HSF4 were excluded due to low expression and relatively narrow SE signal, respectively. The six remaining TFs, namely HOXB2, LBX1, NHLH1, NR2F1 (also known as COUP-TFI), NR2F2 (also known as COUP-TFII) and SOX4 were found to harbor the strongest accessibility and acetylation signals and, therefore, were selected for functional analysis as novel candidate regulators of mDAN differentiation (Figure 3C, Figure EV2).”

Expanded View Figure 2
A)

3. In supplementary 3, in the IPA analysis some data appear with the warning "#iNUM!" at the z-score. Some explanation should be given and if pertinent, added to the table legend.

The term is produced when IPA cannot predict the Z-score and it is represented in our bar graphs in grey (see Figures 6B and 7B). We have added the information to the table header.

4. In methodology, some reagents and techniques appear with a code reference to catalog number and others don't. Please keep it uniform throughout the text.

We have unified the referencing to reagents throughout the Methods.

5. Supplementary table 1 has some TFs highlighted in yellow but there is no legend that explain what the yellow highlight symbolizes. Clarification is needed

This is an error and there should not be any TF highlighted in yellow. We apologize for the inconvenience. The highlights have been removed in the revised tables.

Format suggestions:

1. For an easier to follow flow between figure 3A and the main text, it would be helpful if NR2F1 and NR2F2 graphs in Figure 3A appeared next to each other or one above the other.

We have improved Figure 3 by arranging NR2F1 and NR2F2 loci on top of each other.

2. Supplementary table 2.

a. Data is presented in a confused way. For example, the Top20_TFs_EPIC-DREM is presented as a list of names without divisions of type node or scoring annotations. It would be more informative and easy to follow if proper labeling and scoring is given within this spreadsheet without the necessity of navigating sup.table1 in parallel.

b. it would be preferable to have an extra sheet showing the comparison between both data sets (SE and EPICDREAM) before providing a final list of relevant TFs.

For each of the 17 SE-controlled TFs, the top rank obtained in EPIC-DREM across all the split node, is now included in the last worksheet of Supplementary Table 2.

Reviewer #2 (Significance (Required)):

-General assessment:

Overall, the manuscript is well written, the research laid out in a clear way, and the experiments well thought. The novelty of this study lays in the combination of epigenomic and transcriptomic data at different time points in specific cells during DAN differentiation and description of new roles in DAN differentiation for two TF: LBX1 or NHLH1.

We thank the Reviewer2 for this assessment and believe the revised manuscript has significantly improved.

-Limitations:

One limitation, than the authors themselves mentioned, is the possibility that promising candidate TFs involved in mDAN differentiation are discarded or not taken in account by the EPIC-DREAM algorithm.

We agree with this limitation and will make all the data available for the larger research community to use for follow-up work. Importantly, the data can be easily used for re-analysis using the same pipeline when improved databases become available.

-Audience:

This manuscript focuses on factors involved in mDAN differentiation which targets a highly specific audience however their multiomic and functional methodology might attract broader audiences looking to apply similar pipelines and/or experimentation in different areas of research.

-My field of expertise:

I am a geneticist and neuroscientist with expertise in molecular biology and epigenomics focused on age related neurodegenerative disorders.

-Recommendation:

I believe the conclusions are supported by the results presented and therefore recommend this paper for publication after addressing some minor points.

References:

Grancharova, T., Gerbin, K.A., Rosenberg, A.B. et al. A comprehensive analysis of gene expression changes in a high replicate and open-source dataset of differentiating hiPSC-derived cardiomyocytes. *Sci Rep* 11, 15845 (2021). <https://doi.org/10.1038/s41598-021-94732-1>

Bressan E., Reed X., Bansal V., et al. The Foundational Data Initiative for Parkinson Disease: Enabling efficient translation from genetic maps to mechanism. *Cell Genomics* 3, 100261 (2023). <https://doi.org/10.1016/j.xgen.2023.100261>

Angelopoulou, E., Paudel, Y. N., & Piperi, C. (2019). miR-124 and Parkinson's disease: A biomarker with therapeutic potential. *Pharmacological Research*, 150. <https://doi.org/10.1016/J.PHRS.2019.104515>

Kamath, T., Abdulraouf, A., Burris, S. J., Langlieb, J., Gazestani, V., Nadaf, N. M., ... Macosko, E. Z. (2022). Single-cell genomic profiling of human dopamine neurons identifies a population that selectively degenerates in Parkinson's disease. *Nature Neuroscience*, 25(5), 588–595. <https://doi.org/10.1038/S41593-022-01061-1>

Saraiva, C., Paiva, J., Santos, T., Ferreira, L., & Bernardino, L. (2016). MicroRNA-124 loaded nanoparticles enhance brain repair in Parkinson's disease. *Journal of Controlled Release : Official Journal of the Controlled Release Society*, 235, 291–305. <https://doi.org/10.1016/J.JCONREL.2016.06.005>

Tranchevent, L. C., Halder, R., & Glaab, E. (2023). Systems level analysis of sex-dependent gene expression changes in Parkinson's disease. *Npj Parkinson's Disease* 2023 9:1, 9(1), 1–16. <https://doi.org/10.1038/s41531-023-00446-8>

Walter, J., Bolognin, S., Poovathingal, S. K., Magni, S., Gérard, D., Antony, P. M. A., ... Schwamborn, J. C. (2021). The Parkinson's-disease-associated mutation LRRK2-G2019S alters dopaminergic differentiation dynamics via NR2F1. *Cell Reports*, 37(3). <https://doi.org/10.1016/J.CELREP.2021.109864>

Yang, Y., Li, Y., Yang, H., Guo, J., & Li, N. (2021). Circulating MicroRNAs and Long Non-coding RNAs as Potential Diagnostic Biomarkers for Parkinson's Disease. *Frontiers in Molecular Neuroscience*, 14, 28. <https://doi.org/10.3389/FNMOL.2021.631553/BIBTEX>

Zhang, F., Yao, Y., Miao, N., Wang, N., Xu, X., & Yang, C. (2022). Neuroprotective effects of microRNA 124 in Parkinson's disease mice. *Archives of Gerontology and Geriatrics*, 99. <https://doi.org/10.1016/J.ARCHGER.2021.104588>

Dear Dr. Sinkkonen,

Thank you for the submission of your revised manuscript. We have now received the enclosed reports from the referees. As you will see, while referee 1 is positive, referee 2 is still rather critical with your revised study. I have re-discussed your work with the EMBO reports team, and we have decided that we can accept it for publication, if all limitations of the study are openly discussed.

Please clearly mention that it remains to be demonstrated whether the identified factors are required for mDAN differentiation specifically, and also whether they are essential for mDAN differentiation in vivo.

There are also a few editorial requests that I would like you to address before we can proceed with the official acceptance of your manuscript:

- Please reduce the number of keywords to 5.
- Please correct the conflict of interest subheading to "Disclosure and Competing Interest Statement".
- Please remove the author credits from the ms file. All credits need to be entered in our online submission system now.
- Please remove "Data not shown" on pages 8, 10 and 14, as per journal policy. Please either show the data or rewrite.
- The FUNDING INFO needs to be part of the Ackn. section; the 2nd paragraph (from "PPMI -..." until "...and Yumanity Therapeutics.") should be inserted in the available box in our online submission system, below More Funders button.
- Tables EV1-EV5 need to be renamed to Datasets EV1-EV5; the callouts and the file names need to be updated accordingly; their legends need to be removed from the Appendix file and provided in each file (as a separate tab/sheet).
- The APPENDIX FILE needs a table of content with page numbers on the first page.
- The synopsis image looks nice, but the text is too small at the final image size of 550 pixels wide x 400 pixels large. Please send us a new image at the exact final size (the current image is too large) with readable text.
- Please also send us a short (1-2 sentences) summary of the findings and their significance, and 2-3 bullet points highlighting key results, for our website.
- The Supporting Information paragraph should be removed from the ms.
- Legends for the EV figures need to be listed in the main ms after the main figure legends.
- Figure 3 is in landscape format and needs to be changed to portrait format.
- The acknowledgments section should be moved to right after the Data Availability section.
- Please address these comments from our data editors:
 1. Please note that a separate 'Data Information' section is required in the legends of figures 4; 5; 6
 2. Please indicate the statistical test used for data analysis in the legends of figures 2b; EV3b
 3. Please define the annotated p values ****/* in the legend of figure EV3b as appropriate.
 4. Please note that the box plots need to be defined in terms of minima, maxima, centre, bounds of box and whiskers, and percentile in the legend of figures 1c; 3c; 6d; EV2; EV3a-b
 5. Please note that information related to n is missing in the legend of figures 3c; 5a; 6a; 7a; EV2; EV3a-b.
 6. Please note that the error bars are not defined in the legend of figures 4b-f; 5c, f-g; 6d-g; 7e-g; EV4a-c.
- In our routine image analysis of accepted ms figures, we noted that some panels in Fig 4D and Appendix Fig 2A and in EV Fig 4A and C and App Fig 2B and D seem to be identical, even though the labels sometimes differ. Can you please clarify and send us the source data for these figures?

I would like to suggest some changes to the title and abstract that needs to be written in present tense. Please let me know whether you agree with the following:

Multomics analysis identifies novel facilitators of human dopaminergic neuron differentiation

Midbrain dopaminergic neurons (mDANs) control voluntary movement, cognition, and reward behavior under physiological conditions and are implicated in human diseases such as Parkinson's disease (PD). Many transcription factors (TFs) controlling human mDAN differentiation during development have been described, but much of the regulatory landscape remains undefined. Using a tyrosine hydroxylase (TH) human [OK?] iPSC reporter line, we here generate time series transcriptomic and epigenomic profiles of purified mDANs during differentiation. Integrative analysis predicts novel regulators of mDAN differentiation and super-enhancers are used to identify key TFs. We find LBX1, NHLH1 and NR2F1/2 to promote mDAN differentiation and show that overexpression of either LBX1 or NHLH1 can improve mDAN specification. A more detailed investigation of TF targets reveals that NHLH1 promotes the induction of neuronal miR-124, LBX1 regulates cholesterol biosynthesis, and NR2F1/2 controls neuronal activity.

Referee #1:

After a careful read of the reviewed version of the manuscript EMBOR-2023-57349V1 by Sinkkonen et al. as well as the author's comments to reviewers and new additions to the text I find that, to my understanding, the manuscript scope and conclusions are supported by the presented data.

In addition, the authors addressed all my previous concerns and clarified doubts clearly Therefore I have no objections and recommend this paper for publication without further revision.

Referee #2:

While the authors have made efforts to address the reviewer's comments, they have yet to provide conclusive evidence demonstrating that the identified markers are pivotal in the direct generation of dopaminergic neurons. For example when LMX1A, FOXA2, and NGN2 are knocked down, dopaminergic neurons are not produced. The three genes overexpression increases dramatically the production of dopaminergic neurons (Sheti et al, 2022). Here, quantification is presented as TH-positive area over number of neurons which does not inform on the total number of neurons TH/MAP2.

It's worth noting that many of the genes identified in the study are also expressed in GABAergic neurons (NR2F1 and NR2F2) and glutamatergic neurons (NHLH1 and NR2F2), according to the Allen Brain Map. Additionally, LB1X has been observed in dorsal spinal cord interneurons (Gross et al, 2002, Muller et al, 2002) and somatosensory neurons in the hindbrain (Sieber et al, 2007). Consequently, these candidate genes may not exclusively specify dopaminergic neurons but might instead serve as general regulators of neuronal differentiation.

All editorial and formatting issues were resolved by the authors.

Dr. Lasse Sinkkonen
University of Luxembourg
Department of Life Sciences and Medicine
6, Avenue du Swing
Belvaux, orcid||||||| L-4367
Luxembourg

Dear Dr. Sinkkonen,

I am very pleased to accept your manuscript for publication in the next available issue of EMBO reports. Thank you for your contribution to our journal.
